

# An improved differential evolution algorithm for multi-modal multi-objective optimization

Dan Qu[1,2], Hualin Xiao[1], Huafei Chen[2] and Hongyi Li[2]

[1] College of Mathematics Education, China West Normal University, Nanchong, China
[2] College of Mathematics and Statistics, Sichuan University of Science & Engineering, Zigong, China

## ABSTRACT

Multi-modal multi-objective problems (MMOPs) have gained much attention during the last decade. These problems have two or more global or local Pareto optimal sets (PSs), some of which map to the same Pareto front (PF). This article presents a new affinity propagation clustering (APC) method based on the Multi-modal multi-objective differential evolution (MMODE) algorithm, called MMODE_AP, for the suit of CEC'2020 benchmark functions. First, two adaptive mutation strategies are adopted to balance exploration and exploitation and improve the diversity in the evolution process. Then, the affinity propagation clustering method is adopted to define the crowding degree in decision space (DS) and objective space (OS). Meanwhile, the non-dominated sorting scheme incorporates a particular crowding distance to truncate the population during the environmental selection process, which can obtain well-distributed solutions in both DS and OS. Moreover, the local PF membership of the solution is defined, and a predefined parameter is introduced to maintain of the local PSs and solutions around the global PS. Finally, the proposed algorithm is implemented on the suit of CEC'2020 benchmark functions for comparison with some MMODE algorithms. According to the experimental study results, the proposed MMODE_AP algorithm has about 20 better performance results on benchmark functions compared to its competitors in terms of reciprocal of Pareto sets proximity (rPSP), inverted generational distances (IGD) in the decision (IGDX) and objective (IGDF). The proposed algorithm can efficiently achieve the two goals, *i.e.*, the convergence to the true local and global Pareto fronts along with better distributed Pareto solutions on the Pareto fronts.

# INTRODUCTION

Most of optimal decision issues in research and application can be summarized as multi-objective problem (MOP),which is generally formulated as follows (*Rao, 1991*):

$$min \ F(x) = (f_1(x), f_2(x), \ldots, f_m(x))^T \tag{1}$$

where $x = (x_1, x_2, \ldots, x_n)^T \in \Omega$ is the decision vector of dimension $n$ and $\Omega \in R^n$ is the feasible space. The image set, $S = \{F(x)|x \in \Omega\}$, is called OS. The feasible

Corresponding author
Hualin Xiao,
hualin_xiao688@163.com

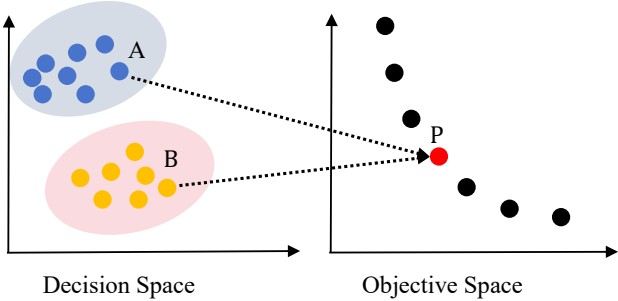

Decision Space          Objective Space

**Figure 1** **Illustration of a two-objective MMOP with global Pareto optimal set (PS) and global Pareto front.**

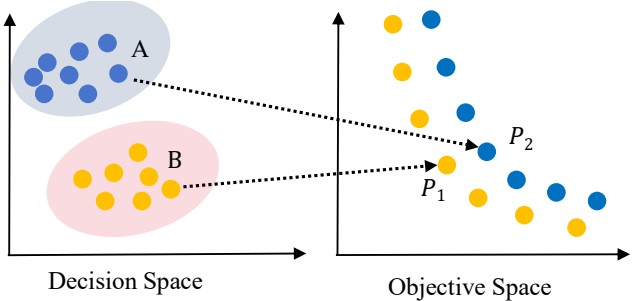

Decision Space          Objective Space

**Figure 2** **Illustration of a two-objective multi-modal multi-objective problem (MMOP) with global and local PS and PF.**

solutions $x_1$ is dominated by $x_2$ on the condition that $f_i(x_1) \leq f_i(x_2), \forall i=1,2,\ldots,m$ and $\exists j \in \{1,2,\ldots,m\}, f_j(x_1) < f_j(x_2)$. Furthermore, if a certain solution is not Pareto dominated by any other solutions, it is called a non-dominated solution. The set of all non-dominated solutions constitutes a Pareto optimal set (PS) and its image in objective space (OS) forms the Pareto front.

In recent years, the study on multi-objective evolutionary algorithms (MOEAs) (*Deb, 2001*) is in the development stage. Many efficient algorithms have been proposed, such as non-dominated sorting based genetic algorithm (NSGA-II) (*Deb et al., 2002*), multi-objective evolutionary algorithm based on decomposition (MOEA/D) (*Zhang & Li, 2008*), $\varepsilon$-dominance based multi-objective evolutionary algorithm ($\varepsilon$-MOEA) (*Deb, Mohan & Mishra, 2005*), improving the Strength Pareto Evolutionary Algorithm for Multi-objective Optimization (SPEA2) (*Zitzler, Laumanns & Thiele, 2001*). However, these MOEAs focused on finding the Pareto Front in OS, ignoring the convergence and diversity in decision space (DS). So, these algorithms are inefficient in solving MMOPs, which have multiple PSs in DS mapping to the same Pareto front in OS as shown in Figs. 1 and 2. Figure 1 depicts an

MMOP example, illustrating that the distance between two points is large in the DS while mapping to the same objective vector.

Many problems in the real world have exhibited multi-modal characteristics, *e.g.*, feature selection problems (*Yue et al., 2019*), the design of space missions (*Schutze, Vasile & Coello, 2011*) and imaging problems (*Sebag et al., 2005*). Obtaining all optimal solutions is necessary for decision-makers (DMs) to understand the problem entirely. Furthermore, it can be affected by some constraints that transfer to another solution. Therefore, decision-makers hope to provide equivalent and diverse solutions for making decisions when dealing with MOPs.

To address MMOPs, many researchers have proposed different multi-modal multi-objective optimization algorithms (*Qu, Suganthan & Liang, 2012*; *Li et al., 2017*; *Shi et al., 2019*; *Maity, Sengupta & Sha, 2019*; *Li, 2010*) and have shown good performance on benchmark problems. The most representative is the omni-optimizer proposed by *Deb & Tiwari (2005)*, which extended NSGAII with several techniques, including Latin hypercube sampling-based population, restricted mating selection and alternative crowding distance in both DS and OS. In 2016, *Liang, Yue & Qu (2016)* proposed a modified NSGAII called DN_NSGAII which adopted the crowding distance (CD) mechanism for forming the mating pool to enhance the diversity in the DS. Additionally, the CD method was used to maintain multiple PSs in the DS. However, the CD method is only considered in the DS without considering the OS. To address the limitations of existing multi-modal multi-objective particle swarm optimization techniques, *Yue, Qu & Liang (2018)* proposed a MO_Ring_PSO_SCD method, which utilized ring-topology to construct a stable niching to enhance the diversity. Moreover, a comprehensive special crowding distance (SCD) is designed to describe the crowding degrees considering both DS and OS. Then, *Liu, Yen & Gong (2018)* adopted two archive and recombination strategies in TriMOEA-TA&R to solve MMOP. *Zhang et al. (2019)* designed a new algorithm, using the clustering method to search for PSs and maintain the diversity. The proposed algorithm enhance the balance between exploration and exploitation.

Recently, numerous multi-objective multi-modal evolutionary algorithms (MMEAs) have been introduced for solving MMOPs. In 2021, *Liang et al. (2021a)* combined the clustering method and elite selection mechanism in MMODE. Clustering-based special crowding distance (CSCD) method is designed to obtain a well-distributed population in DS and OS. *Kahraman et al. (2022)* proposed an improvement of MOAGDE (*Duman, Akbel & Kahraman, 2021*) called DSC-MOAGDE. In this method, three spaces were created and strategies using different reference spaces were designed, which balances the exploitation-exploration successfully and avoids the issue of premature convergence in both spaces. The DSC method has been proven to be an effective solution to overcome local solution traps. In 2022, *Gao et al. (2022)* proposed a decomposition-based multi-modal multi-objective evolutionary algorithm MOEA/D-SS, which adopts a density-based estimation strategy for estimating the number of PSs. MOEA/D-SS can faithfully locate all PSs more accurately. The environmental selection is developed to adjust sub-population size to maintain the diversity dynamically. To solve the problem of premature convergence, *Keivanian & Chiong (2022)* designed an enhanced ICA variant MOFAEICA to tackle MMOPs that

adopted two parallel fuzzy reasoning mechanisms. A novel penalized sigma diversity index is developed for estimating the diversity of solutions in the same rank during Pareto front approximation. *Qu et al. (2022)* proposed a grid-guided multi-objective particle swarm optimizer (GG-MOPSO), which adopted the grid-guided technique to improve the diversity and search efficiency.

Although there are many MMEAs, most works have the same drawbacks. Most MMEAs only concentrate on finding a set of trade-off solutions in objective space instead of finding multiple equivalent solution vectors in decision space. Therefore, balancing the diversity between DS and OS seems to be a challenge for them. This article proposes a novel MMEA to deal with these challenges. Further details are provided in the next section.

The main contributions of this work can be enumerated as follows:

(1) MMODE_AP: The differential evolution algorithm, termed as MMODE_AP, to solve MMOPs is proposed. The affinity propagation clustering technique is designed to group individuals in the same non-dominated layer into different groups. Based on this approach, it is more likely to promote algorithm convergence in the optimization evolution process.

(2) Mutation strategy: Two mutation strategies are customized to bring about priorities in the solution generation process. Based on the strategy, the exploration and mining performance are adaptively balanced and the diversity in DS and OS are further improved.

(3) Comprehensive crowding degree: The CSCD measures the comprehensive crowding distance of solutions to balance diversity and convergence. The traditional crowding distance method can be improved by using the comprehensive crowding distance to reflect the real crowding degree better.

(4) Archive updating: the archive updating approach is designed to balance the convergence and diversity quality of solutions, in which the quality of the global and local PSs is controlled by a parameter that can be set by the user. Based on this, the algorithm can simultaneously identify global and local PSs with high convergence and diversity. Specifically, when $\varepsilon = inf$, the MMODE_AP degrades into a regular MMEA that concentrates on global PSs.

## PRELIMINARIES OF STUDY

MODE is widely used for solving MMOPs, which are named as MMEAs. First, the definition of DE algorithm and APC is presented. The corresponding definitions, including local PS, global PS, local PF and global PF, can be referred to *Li et al. (2022)*. In a typical MMOP, only one global PF maps to either a single PS or multiple global PSs, as shown in Fig. 1. However, in an MMOP with a local Pareto front (MMOPL), there exist local PFs and only one global PF, as depicted in Fig. 2.

### Related studies

The literature shows that numerous researches have been carried out on (i) crowding distance method, (ii) mutation operator design, and (iii) environmental selection for better PF sets.

First, several crowding distance-based methods for selecting non-dominated solutions have been introduced in the literature. In NSGA-II, *Deb et al. (2002)* designed the crowding distance method to truncate the population and find PFs and PSs effectively. As mentioned in *Kahraman et al. (2022)*, DS, OS and unified space are the main reference spaces in the crowding distance approach. Therefore, there are three cases to be discussed.

The first case only references the DS. For example, *Liang, Yue & Qu (2016)* proposed a modified NSGAII called DN_NSGAII, which employs the CD mechanism in DS. *Wu a, Gong & Wang (2021)* introduced a k-means and species-based algorithm (KSDE), which combines k-means clustering and species-based optimization techniques. In KSDE, clustering separates the population into sub-regions based on spatial positional relationships. The crowding factor in KSDE was equal to the population size. A few of these include evolutionary multi-objective optimization algorithm with a decomposition strategy in the decision space (EMO-DD) (*Yang et al., 2021*), a prediction strategy based on decision variable analysis (DVA) (*Zheng et al., 2021*), improved artificial electric field algorithm for multi-objective optimization (*Petwal & Rani, 2020*), and the evolutionary algorithm using a convergence penalized density method (CPDEA) (*Liu et al., 2020*).

The second case only references the OS. For example, *Zadeh, Sayadi & Kosari (2019)* presents the EMOPSO algorithm, an efficient meta-model-based collaborative optimization architecture for solving high fidelity multi-objective multidisciplinary design optimization problems. Star topology was employed in the Pareto-based crowding distance method. Aiming at the MOP with uncertain parameters, *Huang et al. (2022)* designed an outlier removal (OR) mechanism and a new crowding distance in the NSGA-II, the negative effects of parameter uncertainties are reduced. The new method, OR-NSGA-II, can adapt to the MOP with parameter uncertainties. A few of these include a reference-point-based many-objective evolutionary algorithm (NSGA-III) (*Deb & Himanshu, 2014*), the multi-objective equilibrium optimizer with exploration exploitation dominance (MOEO-EED) strategy (*Abdel-Basset et al., 2021*), multi-constraint multi-objective evolutionary algorithm (MC/MOEA) (*Liagkouras & Metaxiotis, 2021*), and guided population archive whale optimization algorithm (GPAWOA) (*Got, Moussaoui & Zouache, 2020*).

The third case references both the OS and DS. For example, in the omni-optimizer algorithm (*Deb & Tiwari, 2005*), alternative crowding distances in both DS and OS are embedded to extend NSGAII. *Yue, Qu & Liang (2018)* modified the crowding distance in the omni-optimizer and proposed MO_Ring_PSO_SCD, which referenced both DS and OS adopting ring-topology to construct the stable niching for better explore ability. In this method, a comprehensive SCD is designed to describe the crowding degrees considering both DS and OS. *Liang et al. (2021a)* designed a CSCD method calculating the comprehensive crowding degree to achieve a evenly distributed population in both DS and OS. A few of these include a novel MMOEA with dual clustering in the decision and objective spaces (MMOEA/DC) (*Lin et al., 2021*), the Multi-modal Multi-objective Differential Evolution Algorithm Using Improved Crowding Distance (MMODE_ICD) (*Yue et al., 2021*), an evolutionary algorithm using a convergence-penalized density method (CPDEA) (*Liu et al., 2020*) and a novel multi-modal multi-objective differential evolution algorithm (MMO_CDE) (*Zhou et al., 216*).

Since the crowding-distance calculation is performed in the unified space, both the DS and OS simultaneously affect the density. Recently, *Kahraman et al. (2022)* proposed a crowding distance-based archive handling approach using unified space.

Second, studies are performed to balance exploitation and exploration capabilities by designing mutation operators. *Yue et al. (2021)* utilized a differential vector generation approach during the variation step to balance exploration and exploitation, thereby enhancing the diversity in both DS and OS. *Javadi, Zille & Mostaghim (2021)* employed weighted sum crowding distance and a neighborhood mutation operator to improve the convergence performance of the NSGA-II. *Zhou et al. (216)* proposed a neighborhood-based dual mutation operator by developing two novel mutation operators, which provide a better balance between exploration and exploitation in locating multiple equivalent PSs. *Liang et al. (2021b)* designed a mutation strategy based on variable classification and elite individuals to improve the performance of offspring by using two different mutation strategies. To avoid premature convergence, *Petwal & Rani (2020)* combined bounded exponential crossover with polynomial mutation. *Abdel-Basset et al. (2021)* adopted the Gaussian method to improve mutation strategy.

Third, numerous recent works have been conducted on the environmental selection process. *Wei et al. (2022)* designed the environmental selection strategy by combining non-dominated sorting and hierarchical clustering to choose promising solutions for the next generation. *Zhou et al. (216)* developed a neighborhood-based dual mutation (NDM) strategy and a clustering-based environmental selection (CES) mechanism to maintain the convergence and diversity of Pareto optimal solutions. *Gao et al. (2022)* combined the estimation strategy and the greedy selection in the environmental selection phase to maintain the population diversity. *Das et al. (2023)* employed an improved environmental selection strategy to pick out high-quality solutions and balance convergence and diversity in solving MOPs. *Li et al. (2021)* designed an environmental selection strategy that designed an operator that removes inefficient solutions to maintain good convergence and diversity of PSs. *Zhang et al. (2022)* proposed a two-stage environmental selection strategy to obtain better convergence of the OS and the distribution of the DS. In addition, several successful studies on environmental selection mechanisms are a novel constrained many-objective optimization EA with enhanced mating and environmental selections (CMME) *Ming et al. (2022)*, a dynamic multi-objective optimal evolutionary algorithm which is based on environmental selection and transfer learning (DMOEA-ESTL) *He, Zheng & Peng (2022)*, an MOEA with a flexible reference point for clustering (MOEA/FC) *Liu et al. (2021)*, and a Multi-modal Multi-Objective differential evolution algorithm using the clustering-based special crowding distance method and elite selection mechanism (MMODE_CSCD) (*Liang et al., 2021a*).

To summarize, to identify global and local PSs that meet user's preferences in resolving MMOPs, it is essential to carry out intensive research on the crowding-distance calculation, mutation operator design and environmental selection in MMEAs. However, compared to recent studies, there is no universally recognized approach for the above three procedures. We are confident that the approach introduced in this study can address the current gap effectively.

## Differential evolution

Differential evolution (DE) is a simple and powerful evolutionary algorithm introduced by *Price & Storn (1997)*. DE uses a series of operations to generate offspring, such as mutation and crossover of current population individuals. DE algorithm also has lower spatial complexity. Because of its excellent characteristics, the DE algorithm is more conducive to the processing of large-scale, high-complexity optimization problems in many fields, including computer science, biology, and industry (*Cappiello et al.,* ; *Bano, Bashir & Younas, 2020*; *Ibrahim et al., 2015*; *Taha et al., 2023*; *Chen et al., 2022*; *Zhang, Yu & Wu, 2021*). The classical DE algorithm (*Storn & Price, 1997*) contains four main steps during the optimization process. The algorithmic details of the four steps are outlined in Algorithm 1.

---
**Algorithm 1 Outline of DE's main procedure**

---
Step 1. Initialize population $P$ randomly and evaluate the individuals of $P$.

Step 2. While stopping criterion is not be met, do

    2.1 for $P_i$, i $=1$, $2, \ldots, NP$ in $P$ repeat:

    (2.1.1) Create candidate from parent $P_i$

    (2.1.2) Evaluate the candidate

    (2.1.3) If the candidate is better, the candidate replaces the parent. Otherwise, the discarded the candidate.

    2.2 Randomly enumerate the individuals in $P$.

## Affinity propagation

The clustering method (*Marques & Wu, 2002*; *Frey & Dueck,* ) divides objects into groups or clusters according to their similarities in some attributes. The general description of clustering analysis is as follows:

Let $X = \{x_1, x_2, \ldots, x_N\} \in R^n$, $x_i$ is a vector of $X$. Clustering is the process of dividing $X$ into m non-empty subsets according to some criteria, which satisfies the following three conditions:

$$C_i \neq \varnothing, i = 1, 2, .., m \tag{2}$$

$$\cup_{i=1}^{m} C_i = X \tag{3}$$

$$C_i \cap Cj = \varnothing, i \neq j, i, j = 1, 2, \ldots, m. \tag{4}$$

The vectors contained in $C_i$ are more similar to each other. The $C_i$ are called clusters of $X$. The typical clustering process includes the following steps: feature selection, proximity measure, clustering criterion, clustering algorithm, and result validation. The similarity matrix is denoted by $S_{N \times N}$, and the similarity is measured by the Euclidean distance between two data points. The elements on the diagonal line of the matrix $S(i, i)$ are replaced by $p(k), k = 1, 2, \ldots, N$, which affects the final number of clusters. The larger value of $p(k)$ is associated with the greater probability that this data point will eventually become a cluster. Therefore, $P = [p(1), p(2), \ldots, p(N)]$ is called the reference degree.

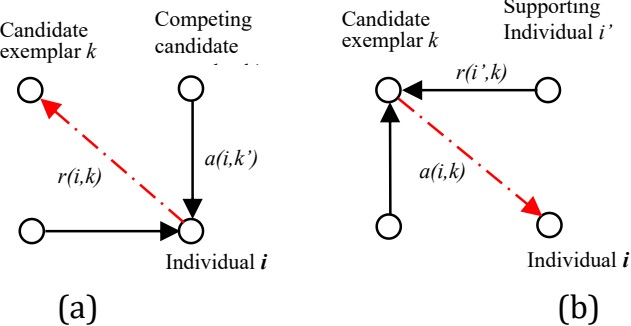

**Figure 3** **Message-passing in APC.** (A) Sending responsibilities. (B) Sending availabilities.

In the affinity propagation algorithm, responsibility and availability are two significant factors to indicate information between points; the former is denoted as responsibility $r(i, k)$, and the latter is availability $a(i, k)$. The responsibility $r(i, k)$ reflects the accumulated evidence for how suitable point $x_k$ is to serve as the exemplar for point $x_i$, which is illustrated in Fig. 3A. The availability $a(i, k)$ reflects the accumulated evidence for how appropriate it would be for point $x_i$ to choose point $x_k$ as its exemplar, as shown in Fig. 3B. In practice, the cyclic calculation process of AP algorithm is the process of exchanging massage of matrix $A = [a(i, k)]_{N \times N}$ and matrix $R = [r(i, k)]_{N \times N}$. Initialize the factor $a(i, k) = 0$. Then, update the two factors with the following formula:

$$r(i, k) = s(i, k) - max_{k \neq k'}[a(i, k') + s(i, k')] \tag{5}$$

$$a(i, k) = \begin{cases} min\left\{0, r(k, k) + \sum_{i' \neq i}[0, r(i', k)]\right\}, i \neq k \\ \sum_{i' \neq i} max[0, r(i', k)], i = k \end{cases} \tag{6}$$

For the point $x_i$, the value of $k$ that maximizes $a(i, k) + r(i, k)$ can identify point $x_i$ as an exemplar if $k = i$, otherwise point $x_i$ belongs to the cluster that $k$ is the exemplar. The AP algorithm implements an iterative process searches for clusters until a high-quality set of exemplars and corresponding clusters are assembled. The procedure of the AP algorithm is depicted in the following Algorithm 2:

# PROPOSED METHOD

## Framework of MMODE_AP

The procedure and flow chart of the MMODE_AP algorithm are depicted in Algorithm 3 and Fig. 4, respectively:

## Solution generation

Since the differential evolution operator (*Price, Storn & Lampinen, 2005*) usually outperforms other mutation operators in single-objective optimization, the MMODE

---

**Algorithm 2 Outline of AP's main procedure**

---

Input: DATA: points to be clustered, $G_{max}$: maximum iterative generations

Output: cluster which each data point belongs

Step 1. Initialization: generation counter $t = 0$; reference degree $p(k)$; similarity matrix $S$;

Step 2. main loop:

    2.1 Calculate matrix $A = [a(i,k)]_{N \times N}$ and $R = [r(i,k)]_{N \times N}$

        2.2 Update the matrix $A$ and $R$ according to the formula:

$$r^t(i,k) = (1 - \lambda) * r^t(i,k) + \lambda * r^{t-1}(i,k)$$
$$a^t(i,k) = (1 - \lambda) * a^t(i,k) + \lambda * a^{t-1}(i,k)$$

    2.3 Determine the center of the points

        2.4 Distribution of each point

        2.5 $t = t + 1$

Step 3. Stop iteration: if $t > G_{max}$, print out the result , otherwise turn to Step 2.

---

**Algorithm 3 Main steps of  MMODE_AP**

---

Input: MOPs: Multi-objective optimization problems; $n$: population size ; $G_{max}$: Maximum Generation;

Output: $A$: Archive

1. Generation counter $t = 1$.

2. Initialization population $P_t = \{x_1, x_2, \ldots, x_{NP}\}$

3. **While** $t < G_{max}$

4. Set external archive $A_t = \varnothing$.

5.  **Non-dominated sorting** on $P_t$ to obtain $P*$

6. For each $x_i \in P_t, i = 1, 2, \ldots, n$ identify an exemplar

7. Generate a new solution offspring $OP_t = $ **solution generation** ( $x_i$) // Algorithm 4

8. Preserve the new solution $P' = OP_t \cup P*$

9. Update the population $P_{t+1}$  **environmental selection** ( $P'$) // Algorithm 5

10.  **Update the archive** $A_{t+1}$// Algorithm 6

11.  **End while**

---

algorithm adopts it and uses the following two methods to improve its performance, which is shown in Algorithm 4.

Firstly, adopt the DE/rand/2 strategy or the DE/current-to-exemplar/1 strategy according to the random value. If the random value is less than $p_1$, then select five random solutions from the whole population and utilize the DE/rand/2 strategy to produce a candidate solution. Otherwise, use the DE/current-to-exemplar/1 strategy with the probability $1 - p_1$, where $p_1 = 1 - \frac{G_c - 1}{maxgen}$, $G_c$ refers to the current generation number while *maxgen* refers to the maximum generation number. These two parameters $G_c$ and $\frac{G_c - 1}{maxgen}$ serve the same role of temperature and cooling schedule came from the method of simulated annealing (SA). The original DE does not have such a parameter and in many cases it may be considered

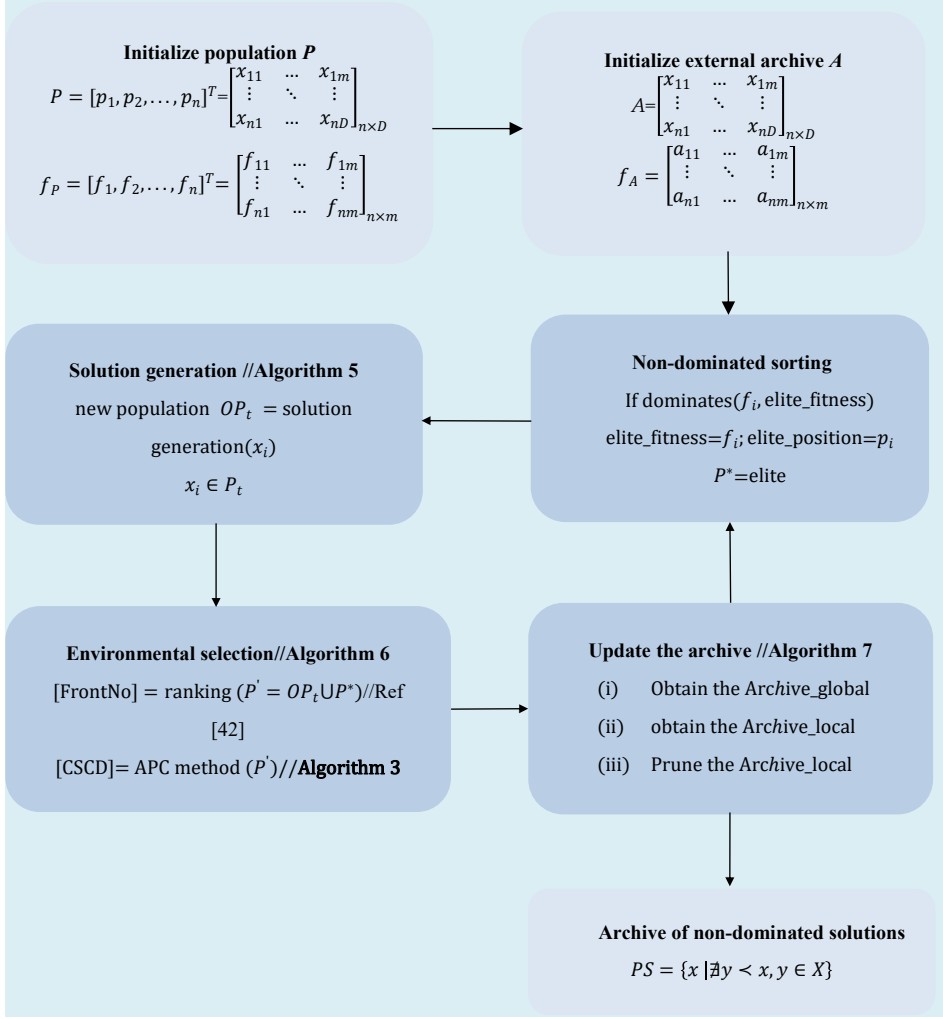

**Figure 4  The flow chart of MMODE_AP.**

an advantage and thus introduction of such a parameter is to be justified together with the formula for $p_1$ as other decreasing functions can be used here similar to SA cooling schedules. Hence, the exploration ability decreases from the initial process of evolution to the end as $p_1$ decreases from 1 to 0. In the proposed selection mechanism, individuals to generate difference vectors are selected in two methods adaptively. The first selection method enhances exploration ability and avoids iterations falling into local optima, the second method helps improve the convergence and diversity in DS and OS. Therefore, a probabilistic-based mutation strategy is used for the mutation. Next, repair the candidate solution to avoid violating the boundary constraints. There are other approaches for handling boundary constraints in DE. *Lampinen (2002)* replaced parameter values that violate boundary constraints with random values generated within the feasible range. *Kozlov, Samsonova & Samsonova (2016)* introduce a new parameter to transform $y_i'$. Here, we adopted one more convenient way to replace parameter values that violate boundary

---

**Algorithm 4 Outline of solution generation**

---

Input: parent solution $x$; adaptive method parameter $p_1$;

Output: candidate solution $y$

**1.** Generate a rand value $r$

**2.** If $r < p_1$

Select five different random solutions $x_{r_1}, x_{r_2}, x_{r_3}, x_{r_4}, x_{r_5}$ from the whole population.

Create a candidate solution use DE/rand/2 strategy: $y_i = x_{r_1} + F * [(x_{r_2} - x_{r_3}) + (x_{r_4} - x_{r_5})]$

Else

Create a candidate solution use DE/current-to-exemplar/1 strategy: $y_i = x_i + F * [(x_{exemplar} - x_i) + (x_{r_1} - x_{r_2})]$

End

**3.** Repair the candidate solution: $y_i' = \begin{cases} a_i & y_i' < a_i \\ b_i & y_i' > a_i \\ y_i' & otherwise \end{cases}$ where $a_i$ and $b_i i = 1(, 2, \ldots, n)$ is the lower and upper boundaries.

**4.** Crossover the candidate solution: $y_i'' = \begin{cases} y_i' & if\ rand_j < C_r\ or\ j = k \\ y_i & otherwise \end{cases}$

where $j = 1, 2, \ldots, n$ and $rand_j \in U[0, 1]$, $C_r$ is crossover rate, $k$ is a random integer in the range of $[1, n]$.

Return $y_i''$.

---

constraints with their upper and lower boundary values. Then, the crossover is used to mutate the candidate to improve its quality further. In this operator, $C_r$ denotes the control parameter. Sometimes, the newly produced candidate may violate the boundary constraints. In such cases, we simply replace them with the closest boundary value. Moreover, this approach does not require the construction of a new candidate. It is worth mentioning that the DE's mutation operation is only included in the MMODE_AP. That is, the selection operation is redefined in the next part. This design's purpose is that the DE mutation operator is invariant of any orthogonal coordinate rotation, which can solve multi-complicated Pareto sets (PSs) (*Li & Zhang, 2009*). The reason why these two mutation strategies are adopted in the solution generation process is that the exploration and exploitation abilities are balanced, and the diversities in DS and OS are improved. In the first strategy, the DE/rand/2 approach is used to enhance the exploration ability and avoid trapping into the local area. In the second strategy, select the one in the first non-dominated front as the exemplar. $x_{exemplar}$ refers to the learning model selected by non-dominated sorting methods, intending to achieve a uniform distribution of solutions on PS. Learning models represent individuals with excellent quality and low density, and we hope to generate more excellent solutions around these model individuals. 'Comprehensive comparison' demonstrates the efficacy of the suggested adaptive individual selection approach.

## Environmental selection

The environmental selection aims to truncate the population to obtain the best *NP* individuals. In MMODE_AP, the truncation, which consists of sorting the individuals with non-dominated sorting and evaluating the individuals with the crowding distance metric proposed in NSGA-II, is employed to create a promising population. Algorithm 5 shows the procedure of environmental selection.

---

**Algorithm 5 Outline of environmental selection**

---

Input: current population $P'$

Output: next generation population $P_{t+1}$

1. Assign the solutions $x \in P'$ to fronts $F_1, F_2, \ldots, F_l$ and calculate the CSCD using APC method// Algorithm 2

2. Add all individuals in $P'$ to $P_{t+1}$

3. If $|P_{t+1}| > NP$ then

4. Iteratively add individuals from $F_i$ to $P_{t+1}$. If $|F_1| + |F_2| + \ldots + |F_{l-1}| = s < N$, $|F_1| + |F_2| + \ldots + |F_l| > N$, the next parent population $P_{t+1}$ is constructed from the member of the sets $F_1, \ldots, F_{l-1}$, and from $N - s$ members of the set $F_l$ according to the CSCD.

5. End if

6. Return $P_{t+1}$.

---

## Archive update

After the environmental selection strategy, the diversity of the global optimal solution is maintained. However, the obtained solutions are not satisfactory regarding their convergence quality. Furthermore, all the local PFs are not completely reversed. The archive, including *Archive_global* and *Archive_local*, is applied to improve the convergence performance and control the quality of local PFs respectively. Notably, a predefined parameter is introduced to control the completeness of the local PSs and PFs.

The steps followed by updating the archive are explained in Algorithm 6. The primary task is to preserve two populations, *Archive_global* and *Archive_local*. First, execute non-dominated sorting on the current Archive $A_t$ and pick up the non-dominated solutions to form *Archive_global*. Then add the remaining solutions into *RemainPop* and delete the solutions in *RemainPop* close to solutions in *Archive_global*. It is generally considered that if a point is close to one point in the DS, they are also likely to be near in the OS. Calculate the distance of all pairs of solutions in the *RemainPop*. Specifically, the Euclidean distance is adopted to measure. Moreover, a predefined parameter $\varepsilon$ is introduced to calculate the Pareto domination relationship between solutions. Specifically, we have the following formula:

$$A_{dis} = \varepsilon * \left( \prod_{i=1}^{n} (x_i^{max} - x_i^{min}) \right)^{\frac{1}{n}} \tag{7}$$

Here, $A_{dis}$ represents the average distance of the DS. In the same layer, $x_j$ is considered the neighbor of $x_i$ if and only if the distance of $x_j$ and $x_i$ is smaller than $A_{dis}$ in the DS. Where $n$

---

**Algorithm 6 Outline of _archive update_**

---

Input: current Archive $A_t$, Archive max size $A_{size}$, predefined parameter $\varepsilon$

Output: next Archive $A_{t+1}$

1. Assign the solutions $x \in A_t$ to fronts $F_1, F_2, \ldots, F_{maxf}, Fno = 1, 2, \ldots, maxf$.

2. Obtain the global _PF Archive_global_ $= A_t(Fno == 1)$

3. Assume _RemainPop_ $= A_t(Fno \sim= 1)$, then the nearest solutions to _Archive_global_ are deleted and calculate the distance of all pair of solutions in the _RemainPop_.

4. Obtain the Pareto domination relationship between pair of solutions of _RemainPop_ according to $\varepsilon$ and calculate the Local PF membership ( _LocalM_ ).

5. Find the index of individual whose _LocalM_ equals zero: $index = A_t(LocalM == 0)$, then obtain _Archive_local_ $= RemainPop(index)$.

6. Merge two types archive: $A_{t+1} = [Archive\_global \cup Archive\_local]$.

7. If $|A_{t+1}| > A_{size}$ then

8. Prune _Archive_local_ by non_domination sorting.

9. End if

10. Return $A_{t+1}$.

---

denotes the dimension of the DS. $x_i^{max}$ and $x_i^{min}$ denote the maximum and minimum values of the $i\_th$ decision variable of current front, respectively. $\varepsilon$ is a domain positive parameter controlling the average number of neighbors. Specifically, the optimal solutions of local PSs can not be dominated by any other solutions in the neighbor solutions. Moreover, the local PF membership is calculated within neighboring solutions. The neighbors of $x_i$ are denoted as $N_x$. If there is no solution in $N_x$ dominates $x_i$, then $x_i$ is a local Pareto optimal solution of current layer of _RemainPop_. This way, the solution in the local PS will not be compared with the solution in the global PSs. Therefore, the local PSs can be found. The local PF membership of $x_i$ is computed as follows:

$$LocalM = \sum_{j=1}^{n_i} B_{i,j} \tag{8}$$

where $B_{i,j} = 1$ if and only if $x_j \prec x_i$ and $B_{i,j} = 0$; otherwise. $n_i$ denotes the number of neighbors of $x_i$ in the set $N_x$. Based on the definition of local PF, the one whose _LocalM_ equals 0 can be considered a local Pareto optimal solution. An appropriate value of $\varepsilon$ can balance the quality of the local PF (if it exists) and enhance the efficiency of finding all local Pareto solutions. It is worth noting that the setting of $\varepsilon$ is dispensable on the prior information about the problem. Actually, $\varepsilon$ can serve as an indicator of the difference between the global and local PFs. If the problem possesses several global and local PS, we suggest setting $\varepsilon \in [0.01, 0.1]$ according to the experiment results. Otherwise, $\varepsilon = inf$ and MMODE_AP transform into a regular MMEA focusing only on global PS in this situation. By setting different values of parameter $\varepsilon$, the local PSs and the solutions surrounding the global PS can be maintained. Notably, if there are acceptable solutions in $A_{t+1}$ ( $|A_{t+1}| \leq A_{size}$), then we return $A_{t+1}$. Otherwise, if we prune _Archive_local_ by non-domination sorting approach until the population size of $A_{t+1}$ is less than the threshold value $A_{size}$. The prune method is consistent with the environmental selection operation.

**Table 1  Features of the CEC 2020 benchmark functions.**

| Function name | F1 | F2 | F3 | F4 | F5 | F6 | N_ops (N_global + N_local) |
|---|---|---|---|---|---|---|---|
| MMF1 | × | × | ✓ | Convex | Nonlinear | × | 2+0 |
| MMF2 | × | × | ✓ | Convex | Nonlinear | × | 2+0 |
| MMF4 | × | × | ✓ | Concave | Nonlinear | × | 2+0 |
| MMF5 | × | × | ✓ | Convex | Nonlinear | × | 2+0 |
| MMF7 | × | × | ✓ | Convex | Nonlinear | × | 2+0 |
| MMF8 | × | × | ✓ | Concave | Nonlinear | × | 2+0 |
| MMF10 | × | × | ✓ | Convex | Linear | × | 1+0 |
| MMF11 | × | × | ✓ | Convex | Linear | ✓ | 1+0 |
| MMF12 | × | × | ✓ | Convex | Linear | ✓ | 1+0 |
| MMF13 | × | × | ✓ | Convex | Nonlinear | ✓ | 1+0 |
| MMF14 | ✓ | ✓ | ✓ | Concave | Linear | ✓ | 2+0 |
| MMF15 | ✓ | ✓ | ✓ | Concave | Linear | ✓ | 1+0 |
| MMF1_e | × | × | ✓ | Convex | Nonlinear | × | 2+0 |
| MMF14_a | ✓ | ✓ | ✓ | Concave | Nonlinear | ✓ | 2+0 |
| MMF15_a | ✓ | ✓ | ✓ | Concave | Nonlinear | ✓ | 1+0 |
| MMF10_l | × | × | ✓ | Convex | Linear | × | 1+1 |
| MMF11_l | × | × | ✓ | Convex | Linear | ✓ | 1+1 |
| MMF12_l | × | × | ✓ | Convex | Linear | ✓ | 1+1 |
| MMF13_l | × | × | ✓ | Convex | Nonlinear | ✓ | 1+1 |
| MMF15_l | ✓ | ✓ | ✓ | Concave | Linear | ✓ | 1+1 |
| MMF15_a_l | ✓ | ✓ | ✓ | Concave | Nonlinear | ✓ | 1+1 |
| MMF16_l 1 | ✓ | ✓ | ✓ | Concave | Linear | ✓ | 2+1 |
| MMF16_l 2 | ✓ | ✓ | ✓ | Concave | Linear | ✓ | 1+2 |
| MMF16_l 3 | ✓ | ✓ | ✓ | Concave | Linear | ✓ | 2+2 |

**Notes.**
F1-F6 are respectively scalable number of variables, scalable number of objectives, Pareto optima known, PF geometry, PS geometry and scalable number of PS. N_ops represents the number of PS to be obtained. N_global and N_local represent the number of global PS and the number of local PSs, respectively.

# THE EXPERIMENTAL STUDY

## Experimental settings

### Benchmark problems

In order to validate the performance of the proposed algorithm, this chapter adopts the CEC'2020 multi-modal multi-objective optimization benchmark suite to test the algorithm. This benchmark suite includes 24 test functions with multiple features, as shown in Table 1. Details about the test suite can be found in the corresponding literature. Consequently, the experimental conditions for MMEAs were used (*Liang et al., 2020*). Experimental run settings are provided below:

- All algorithms run 21 times independently on each benchmark problem.
- The population size is set as *200 * N_ops* and *N_ops* means the number of local and global PS to be obtained for the problem.
- Maximal fitness evaluations (*MaxFEs*) was used as the termination criterion. The *MaxFEs* value was assigned as 10,000 *N_ops*.

- The experimental environment uses 11th Gen Intel(R) Core(TM) i5-11320H @ 3.20 GHz 3.19 GHz; 16 GB memory; Windows 11 operating system. The algorithm is implemented in MATLAB R2015a.

The details for each benchmark problem can be referred to (*Liang et al., 2020*).

### *Performance metrics*

Four performance indicators including rPSP (*Yue et al., 2019*), 1/HV (*Liang et al., 2020*), IGD (*Zhou, Zhang & Jin, 2009*) in DS (IGDX) and IGD in OS (IGDF) are employed to discuss the results of the competing algorithms by *Liang et al. (2020)*. Among these four indicators, rPSP and IGDX measure the extent of spread attained in the obtained optimal solution in DS, while rHV and IGDF are adopted to examine the population distribution in OS. Furthermore, a satisfactory IGDX generally has a beneficial IGDF, while a reasonable IGDF does not mean a favorable IGDX (*Liang et al., 2020*).

### *Competitor algorithms*

In order to validate the effectiveness of the proposed algorithm, the state-of-the-art MMEAs, DSC-MOAGDE (*Kahraman et al., 2022*), MO_PSO_MM (*Liang et al., 2018*), MO_Ring_PSO_SCD (*Yue, Qu & Liang, 2018*), DN-NSGAII (*Liang, Yue & Qu, 2016*), Omni_Opt (*Deb & Tiwari, 2005*), NSGA-II (*Deb et al., 2002*), and SPEA2 (*Zitzler, Laumanns & Thiele, 2001*) are chosen as the comparison algorithms. In order to get statistically sound conclusions in this comparison, the specific parameter settings in each algorithm are consistent with the original papers and the actual reference points for both PFs and PSs are provided in the original papers. Moreover, the Wilcoxon rank-sum test (*Wilcoxon, 1945*) with $p < 0.05$ and the Friedman test (*Friedman, 1939*) are employed to compare MMODE_AP with other competitor algorithms. If MMODE_AP performs better or worse than the competitor algorithm, "$+$" or "$-$" is labeled respectively. In addition, if the two algorithms have no obvious difference, "$\approx$" is labeled.

## Comprehensive comparison

The performance of the proposed algorithm on test suites is analyzed in this section. The statistical metric values (*i.e.,* mean and standard deviation values) are shown in Tables 2–5. In comparison tables, the best index values are highlighted in bold. In addition, the average rankings calculated by the Friedman test are shown in Table 6. Due to article length limitations, the results of a typical approximated set obtained by all algorithms are provided in the Fig. S1.

The superiority of MMODE_AP is depicted in the statistical data in Tables 2–6. Statistical comparison results show that MMODE_AP has the best solution performance for the test problems used in the experiment. As we know, the smaller the value of rPSP and 1/HV, the better the convergence and diversity of obtained solutions. Table 2 shows that NSGAII achieves the best average 1/HV values on ten functions. DSC-MOAGDE and DN-NSGAII rank second and third on 9 and 8 functions, respectively. MMODE_AP obtains good results, with a smaller 1/HV than other competitors, but occasionally worse than DN-NSGAII, DSC-MOAGDE and NSGA-II. For MMF7, MMF11, MMF12 and MMF13 test functions,

**Table 2** Comparison of 1/HV mean and standard deviation obtained by all algorithms for CEC 2020 benchmark problems.

| 1/HV | MMODE_AP | DSC-MOAGDE | MO_PSO_MM | MO_Ring_PSO_SCD | DN-NSGAII | Omni_Opt | NSGAII | SPEA2 |
|---|---|---|---|---|---|---|---|---|
| MMF1 | **1.1437 ± 0.0001** | 1.1446 ± 0.0003(≈) | 1.1438 ± 0.0002(+) | 1.1449 ± 0.0002(+) | 1.1447 ± 0.0005(+) | 1.1440 ± 0.0005(+) | 1.1439 ± 0.0007(≈) | 1.1442 ± 0.0002(+) |
| MMF2 | **1.1495 ± 0.0010** | 1.1719 ± 0.0056(+) | 1.1616 ± 0.0025(+) | 1.1686 ± 0.0026(+) | 1.1608 ± 0.0126(+) | 1.1514 ± 0.0062(≈) | 1.1524 ± 0.0065(≈) | 1.1757 ± 0.0112(+) |
| MMF4 | 1.8473 ± 0.0009 | 1.8529 ± 0.0027(+) | 1.8487 ± 0.0005(+) | 1.8513 ± 0.0010(+) | 1.8488 ± 0.0004(+) | 1.8476 ± 0.0002(+) | **1.8467 ± 0.0001**(≈) | 1.8524 ± 0.0039(+) |
| MMF5 | 1.1437 ± 0.0002 | 1.1446 ± 0.0007(+) | 1.1438 ± 0.0001(≈) | 1.1449 ± 0.0003(+) | 1.1443 ± 0.0005(+) | 1.1438 ± 0.0007(≈) | **1.1434 ± 0.0005**(−) | 1.1444 ± 0.0004(+) |
| MMF7 | **1.1432 ± 0.0001** | 1.1445 ± 0.0003(+) | 1.1434 ± 0.0001(+) | 1.1445 ± 0.0002(+) | 1.1448 ± 0.0005(+) | 1.1438 ± 0.0002(+) | **1.1432 ± 0.0002**(≈) | 1.1440 ± 0.0002(+) |
| MMF8 | 2.3704 ± 0.0014 | 2.3904 ± 0.0070(+) | 2.3719 ± 0.0024(≈) | 2.3909 ± 0.0190(+) | 2.3673 ± 0.0010(−) | 2.3648 ± 0.0003(−) | **2.3634 ± 0.0004**(-) | 2.3851 ± 0.0122(+) |
| MMF10 | 0.0801 ± 0.0035 | **0.0775 ± 0.0003**(≈) | 0.0780 ± 0.0002(≈) | 0.0795 ± 0.0003(≈) | 0.0829 ± 0.0027(+) | 0.0807 ± 0.0032(≈) | 0.0817 ± 0.0031(≈) | 0.0796 ± 0.0029(≈) |
| MMF11 | **0.0689 ± 0.0000** | **0.0689 ± 0.0000**(+) | **0.0689 ± 0.0000**(+) | 0.0690 ± 0.0000(+) | **0.0689 ± 0.0000**(+) | **0.0689 ± 0.0000**(≈) | **0.0689 ± 0.0000**(−) | **0.0689 ± 0.0000**(+) |
| MMF12 | **0.6355 ± 0.0000** | 0.6367 ± 0.0009(+) | 0.6368 ± 0.0006(+) | 0.6394 ± 0.0013(+) | 0.6362 ± 0.0007(+) | 0.6633 ± 0.0627(+) | **0.6355 ± 0.0001**(≈) | 0.6360 ± 0.0002(+) |
| MMF13 | **0.0542 ± 0.0000** | 0.0543 ± 0.0001(+) | 0.0543 ± 0.0000(+) | 0.0544 ± 0.0000(+) | 0.0543 ± 0.0000(+) | 0.0543 ± 0.0000(+) | **0.0542 ± 0.0000**(−) | 0.0543 ± 0.0000(+) |
| MMF14 | 0.3420 ± 0.0219 | 0.3293 ± 0.0008(≈) | 0.3497 ± 0.0157(≈) | 0.3545 ± 0.0151(≈) | **0.3269 ± 0.0088**(−) | 0.3315 ± 0.0088(≈) | 0.3551 ± 0.0092(≈) | 0.3852 ± 0.0598(≈) |
| MMF15 | 0.2509 ± 0.0092 | 0.2344 ± 0.0009(≈) | 0.2451 ± 0.0107(≈) | 0.2435 ± 0.0117(≈) | **0.2314 ± 0.0103**(−) | 0.2421 ± 0.0113(≈) | 0.2402 ± 0.0065(−) | 0.2490 ± 0.0314(≈) |
| MMF1_e | 1.1618 ± 0.0135 | 1.2147 ± 0.0008(+) | **1.1527 ± 0.0012**(≈) | 1.1690 ± 0.0127(≈) | 1.2147 ± 0.1093(≈) | 1.1554 ± 0.0085(≈) | 1.1539 ± 0.0060(≈) | 1.1609 ± 0.0125(≈) |
| MMF14_a | 0.3579 ± 0.0166 | 0.3252 ± 0.0009(≈) | 0.3288 ± 0.0194(−) | 0.3358 ± 0.0272(−) | **0.3179 ± 0.0111**(−) | 0.3312 ± 0.0088(−) | 0.3502 ± 0.0045(≈) | 0.3587 ± 0.0474(≈) |
| MMF15_a | 0.2592 ± 0.0116 | **0.2245 ± 0.0009**(≈) | 0.2408 ± 0.0081(−) | 0.2408 ± 0.0099(−) | 0.2446 ± 0.0157(−) | 0.2425 ± 0.0138(−) | 0.2396 ± 0.0099(−) | 0.2652 ± 0.0244(≈) |
| MMF10_l | 0.1051 ± 0.0050 | **0.1036 ± 0.0003**(≈) | 0.0996 ± 0.0038(−) | 0.1058 ± 0.0037(≈) | 0.1766 ± 0.0102(≈) | 0.1492 ± 0.0118(≈) | 0.1566 ± 0.0106(≈) | 0.2747 ± 0.0347(−) |
| MMF11_l | 0.0690 ± 0.0001 | **0.0688 ± 0.0000**(−) | **0.0688 ± 0.0000**(−) | 0.0689 ± 0.0000(−) | **0.0688 ± 0.0000**(−) | **0.0688 ± 0.0000**(−) | **0.0688 ± 0.0000**(−) | **0.0688 ± 0.0000**(−) |
| MMF12_l | 0.8975 ± 0.1014 | 0.6355 ± 0.0006(−) | 0.6361 ± 0.0002(−) | 0.6371 ± 0.0005(−) | 0.6354 ± 0.0000(−) | 0.6353 ± 0.0000(−) | **0.6352 ± 0.0000**(−) | 0.6354 ± 0.0001(−) |
| MMF13_l | 0.0549 ± 0.0002 | **0.0542 ± 0.0009**(−) | **0.0542 ± 0.0000**(−) | 0.0543 ± 0.0000(−) | **0.0542 ± 0.0000**(−) | **0.0542 ± 0.0000**(−) | **0.0542 ± 0.0000**(−) | **0.0542 ± 0.0000**(−) |
| MMF15_l | 0.2639 ± 0.0125 | **0.2253 ± 0.0009**(−) | 0.2346 ± 0.0050(−) | 0.2315 ± 0.0065(−) | 0.2328 ± 0.0086(−) | 0.2313 ± 0.0041(−) | 0.2372 ± 0.0039(−) | 0.2716 ± 0.0386(≈) |
| MMF15_a_l | 0.2578 ± 0.0092 | **0.2223 ± 0.0009**(−) | 0.2308 ± 0.0076(−) | 0.2357 ± 0.0059(−) | 0.2339 ± 0.0116(−) | 0.2295 ± 0.0040(−) | 0.2361 ± 0.0025(−) | 0.2683 ± 0.0253(≈) |
| MMF16_l 1 | 0.2466 ± 0.0138 | 0.2259 ± 0.0034(−) | 0.2319 ± 0.0049(−) | 0.2328 ± 0.0130(−) | **0.2209 ± 0.0038**(−) | 0.2284 ± 0.0048(−) | 0.2352 ± 0.0020(−) | 0.2440 ± 0.0205(≈) |
| MMF16_l 2 | 0.2591 ± 0.0109 | **0.2250 ± 0.0008**(−) | 0.2308 ± 0.0050(−) | 0.2300 ± 0.0075(−) | 0.2251 ± 0.0040(−) | 0.2295 ± 0.0045(−) | 0.2349 ± 0.0027(−) | 0.2598 ± 0.0165(≈) |
| MMF16_l 3 | 0.2485 ± 0.0080 | 0.2255 ± 0.0009(≈) | 0.2299 ± 0.0032(−) | 0.2302 ± 0.0074(−) | **0.2226 ± 0.0041**(−) | 0.2265 ± 0.0018(−) | 0.2344 ± 0.0030(−) | 0.2530 ± 0.0199(≈) |
| + | | 9/24 | 7/24 | 9/24 | 9/24 | 5/24 | 0/24 | 9/24 |
| ≈ | | 8/24 | 6/24 | 5/24 | 2/24 | 8/24 | 10/24 | 11/24 |
| − | | 7/24 | 11/24 | 10/24 | 13/24 | 11/24 | 14/24 | 4/24 |

**Notes.**

Bold numbers indicate the best result in the row of data.

**Table 3** Comparison of 1/PSP mean and standard deviation obtained by all algorithms for CEC 2020 benchmark problems.

| rpsp | MMODE_AP | DSC-MOAGDE | MO_PSO_MM | MO_Ring_PSO_SCD | DN-NSGAII | Omni_Opt | NSGAII | SPEA2 |
|------|----------|------------|-----------|-----------------|-----------|----------|--------|-------|
| MMF1 | $0.0248 \pm 0.0011$ | $0.0278 \pm 0.0022(+)$ | $\mathbf{0.0237 \pm 0.0012}(\approx)$ | $0.0293 \pm 0.0011(+)$ | $0.0597 \pm 0.0093(+)$ | $0.0498 \pm 0.0087(+)$ | $0.0717 \pm 0.0118(+)$ | $0.0352 \pm 0.0033(+)$ |
| MMF2 | $0.0241 \pm 0.0134$ | $0.0301 \pm 0.0076(\approx)$ | $\mathbf{0.0172 \pm 0.0024}(\approx)$ | $0.0258 \pm 0.0050(+)$ | $0.0554 \pm 0.0204(+)$ | $0.0781 \pm 0.0624(+)$ | $0.0581 \pm 0.0346(+)$ | $0.0535 \pm 0.0291(+)$ |
| MMF4 | $\mathbf{0.0135 \pm 0.0008}$ | $0.0156 \pm 0.0008(+)$ | $\mathbf{0.0135 \pm 0.0009}(\approx)$ | $0.0159 \pm 0.0005(+)$ | $0.0425 \pm 0.0072(+)$ | $0.0487 \pm 0.0123(+)$ | $0.0862 \pm 0.0257(+)$ | $0.0298 \pm 0.0113(+)$ |
| MMF5 | $\mathbf{0.0447 \pm 0.0013}$ | $0.0518 \pm 0.0033(+)$ | $0.0457 \pm 0.0018(\approx)$ | $0.0541 \pm 0.0031(+)$ | $0.1193 \pm 0.0132(+)$ | $0.1151 \pm 0.0103(+)$ | $0.1430 \pm 0.0183(+)$ | $0.0678 \pm 0.0092(+)$ |
| MMF7 | $0.0156 \pm 0.0027$ | $0.0155 \pm 0.0005(\approx)$ | $\mathbf{0.0124 \pm 0.0004}(-)$ | $0.0158 \pm 0.0006(\approx)$ | $0.0269 \pm 0.0061(+)$ | $0.0243 \pm 0.0043(+)$ | $0.0476 \pm 0.0080(+)$ | $0.0216 \pm 0.0061(+)$ |
| MMF8 | $0.0349 \pm 0.0021$ | $0.0600 \pm 0.0117(+)$ | $\mathbf{0.0332 \pm 0.0017}(-)$ | $0.0414 \pm 0.0025(+)$ | $0.1201 \pm 0.0384(+)$ | $0.1295 \pm 0.0394(+)$ | $1.6484 \pm 1.8638(+)$ | $1.2735 \pm 1.3402(+)$ |
| MMF10 | $0.0959 \pm 0.1454$ | $0.0201 \pm 0.0052(\approx)$ | $\mathbf{0.0045 \pm 0.0023}(\approx)$ | $0.0239 \pm 0.0050(\approx)$ | $0.1986 \pm 0.1186(+)$ | $0.1177 \pm 0.1256(\approx)$ | $0.1592 \pm 0.1411(+)$ | $0.0802 \pm 0.1250(+)$ |
| MMF11 | $0.0042 \pm 0.0003$ | $0.0042 \pm 0.0003(+)$ | $0.0038 \pm 0.0002(-)$ | $0.0054 \pm 0.0003(+)$ | $0.0045 \pm 0.0003(\approx)$ | $0.0043 \pm 0.0002(\approx)$ | $0.0034 \pm 0.0004(-)$ | $\mathbf{0.0033 \pm 0.0006}(-)$ |
| MMF12 | $0.0018 \pm 0.0001$ | $0.0033 \pm 0.0004(+)$ | $0.0023 \pm 0.0004(+)$ | $0.0041 \pm 0.0006(+)$ | $0.0022 \pm 0.0002(+)$ | $0.0075 \pm 0.0124(+)$ | $0.0018 \pm 0.0004(\approx)$ | $\mathbf{0.0017 \pm 0.0004}(\approx)$ |
| MMF13 | $\mathbf{0.0272 \pm 0.0011}$ | $0.0327 \pm 0.0015(+)$ | $0.0297 \pm 0.0014(+)$ | $0.0355 \pm 0.0011(+)$ | $0.0710 \pm 0.0088(+)$ | $0.0695 \pm 0.0101(+)$ | $0.1248 \pm 0.0781(+)$ | $0.1518 \pm 0.1098(+)$ |
| MMF14 | $0.0465 \pm 0.0022$ | $\mathbf{0.0452 \pm 0.0009}(+)$ | $0.0473 \pm 0.0017(\approx)$ | $0.0463 \pm 0.0015(\approx)$ | $0.0882 \pm 0.0097(+)$ | $0.0798 \pm 0.0039(+)$ | $0.0949 \pm 0.0081(+)$ | $0.2363 \pm 0.0529(+)$ |
| MMF15 | $0.0508 \pm 0.0014$ | $\mathbf{0.0485 \pm 0.0006}(\approx)$ | $0.0497 \pm 0.0009(\approx)$ | $0.0519 \pm 0.0025(\approx)$ | $0.0797 \pm 0.0055(+)$ | $0.0674 \pm 0.0043(+)$ | $0.0796 \pm 0.0136(+)$ | $0.1068 \pm 0.0215(+)$ |
| MMF1_e | $0.9836 \pm 0.6899$ | $0.4986 \pm 0.0519(-)$ | $\mathbf{0.3027 \pm 0.0628}(-)$ | $0.3954 \pm 0.1566(\approx)$ | $0.9277 \pm 0.2830(\approx)$ | $2.5675 \pm 1.8265(+)$ | $1.6827 \pm 0.9162(\approx)$ | $4.6551 \pm 2.2512(+)$ |
| MMF14_a | $0.0531 \pm 0.0019$ | $0.0545 \pm 0.0019(+)$ | $\mathbf{0.0517 \pm 0.0016}(\approx)$ | $0.0527 \pm 0.0014(+)$ | $0.1057 \pm 0.0051(+)$ | $0.0977 \pm 0.0045(+)$ | $0.1189 \pm 0.0133(+)$ | $0.3918 \pm 0.1910(+)$ |
| MMF15_a | $0.0598 \pm 0.0034$ | $0.0574 \pm 0.0033(\approx)$ | $\mathbf{0.0549 \pm 0.0020}(-)$ | $0.0560 \pm 0.0014(-)$ | $0.1073 \pm 0.0092(+)$ | $0.0911 \pm 0.0081(+)$ | $0.1053 \pm 0.0121(+)$ | $0.1994 \pm 0.0497(+)$ |
| MMF10_$l$ | $\mathbf{0.0632 \pm 0.0378}$ | $5.162 \pm 0.0002(+)$ | $0.6252 \pm 0.9481(+)$ | $0.1723 \pm 0.0066(+)$ | $4.4681 \pm 3.4762(+)$ | $3.3254 \pm 3.3817(+)$ | $6.3526 \pm 5.3786(+)$ | $8.2239 \pm 3.3197(+)$ |
| MMF11_$l$ | $\mathbf{0.0690 \pm 0.0452}$ | $0.9266 \pm 0.7406(+)$ | $1.5151 \pm 0.5473(+)$ | $0.4573 \pm 0.4560(+)$ | $1.9395 \pm 0.1703(+)$ | $2.1166 \pm 0.1041(+)$ | $2.9124 \pm 0.8436(+)$ | $3.6882 \pm 1.3752(+)$ |
| MMF12_$l$ | $\mathbf{0.0203 \pm 0.0144}$ | $2.3497 \pm 0.1521(+)$ | $1.2775 \pm 0.5153(+)$ | $0.8244 \pm 0.5110(+)$ | $2.4742 \pm 0.2300(+)$ | $2.6015 \pm 0.1466(+)$ | $6.6292 \pm 4.7300(+)$ | $4.1364 \pm 1.9745(+)$ |
| MMF13_$l$ | $\mathbf{0.0876 \pm 0.0128}$ | $0.3635 \pm 0.0939(+)$ | $0.5017 \pm 0.0700(+)$ | $0.2956 \pm 0.0425(+)$ | $0.5906 \pm 0.0203(+)$ | $0.5856 \pm 0.0134(+)$ | $0.6898 \pm 0.1075(+)$ | $0.7406 \pm 0.1775(+)$ |
| MMF15_$l$ | $\mathbf{0.0822 \pm 0.0075}$ | $0.1557 \pm 0.0156(+)$ | $0.1534 \pm 0.0211(+)$ | $0.1468 \pm 0.0080(+)$ | $0.3163 \pm 0.1312(+)$ | $0.3479 \pm 0.1303(+)$ | $0.4381 \pm 0.1780(+)$ | $0.3791 \pm 0.1041(+)$ |
| MMF15_a_$l$ | $\mathbf{0.1023 \pm 0.0121}$ | $0.1706 \pm 0.0112(+)$ | $0.1625 \pm 0.0132(+)$ | $0.1609 \pm 0.0088(+)$ | $0.2180 \pm 0.0356(+)$ | $0.2431 \pm 0.0355(+)$ | $0.2665 \pm 0.0329(+)$ | $0.2776 \pm 0.0368(+)$ |
| MMF16_$l$ 1 | $\mathbf{0.0755 \pm 0.0060}$ | $0.1182 \pm 0.0008(+)$ | $0.1101 \pm 0.0083(+)$ | $0.1034 \pm 0.0063(+)$ | $0.1974 \pm 0.0302(+)$ | $0.2053 \pm 0.0290(+)$ | $0.1996 \pm 0.0334(+)$ | $0.2269 \pm 0.0345(+)$ |
| MMF16_$l$ 2 | $\mathbf{0.1018 \pm 0.0123}$ | $0.1954 \pm 0.0262(+)$ | $0.2035 \pm 0.0308(+)$ | $0.1852 \pm 0.0131(+)$ | $0.4457 \pm 0.1234(+)$ | $0.4571 \pm 0.1632(+)$ | $0.6643 \pm 0.2413(+)$ | $0.4667 \pm 0.1873(+)$ |
| MMF16_$l$ 3 | $\mathbf{0.0885 \pm 0.0073}$ | $0.1391 \pm 0.0075(+)$ | $0.1470 \pm 0.0139(+)$ | $0.1392 \pm 0.0133(+)$ | $0.2738 \pm 0.0487(+)$ | $0.2653 \pm 0.0411(+)$ | $0.2887 \pm 0.0500(+)$ | $0.2716 \pm 0.0666(+)$ |
| $+$ | | 18/24 | 11/24 | 17/24 | 22/24 | 22/24 | 21/24 | 22/24 |
| $\approx$ | | 5/24 | 8/24 | 6/24 | 2/24 | 2/24 | 2/24 | 1/24 |
| $-$ | | 1/24 | 5/24 | 1/24 | 0/24 | 0/24 | 1/24 | 1/24 |

**Notes.**
Bold numbers indicate the best result in the row of data.

MMODE_AP yield equal mean 1/HV values compared with the winner algorithms. Based on archive update and environmental selection mechanism, MMODE_AP achieves an excellent approximation to the true PF on most of these MMOPs. However, it is difficult to obtain a better approximation to the true PS and true PF simultaneously. That is, the number of the best mean 1/HV values obtained by MMODE_AP is less than other algorithms. The reason might be that MMODE_AP is focused on solving MMOP with the local Pareto front. Table 3 shows that MMODE_AP achieves competitive performance compared to its competitors regarding rPSP. Especially, its performance on functions with multiple PFs is outstanding. Accordingly, the results show that the performance of MMODE_AP whose framework embedding the AP approach is excellent in terms of convergence. Compared with DN-NSGAII and NSGA-II, MMODE_AP has a narrow advantage in some instances. Table 4 shows that MMODE_AP obtains better results, with the smaller IGDX compared with some state-of-the-art algorithms on the test

**Table 4  Comparison of IGDX mean and standard deviation obtained by all algorithms for CEC 2020 benchmark problems.**

| IGDX | MMODE_AP | DSC-MOAGDE | MO_PSO_MM | MO_Ring_PSO_SCD | DN-NSGAII | Omni_Opt | NSGAII | SPEA2 |
|---|---|---|---|---|---|---|---|---|
| MMF1 | 0.0247 ± 0.0011 | 0.0274 ± 0.0021(+) | **0.0236 ± 0.0012**(-) | 0.0292 ± 0.0010(+) | 0.0593 ± 0.0092 (+) | 0.0494 ± 0.0087(+) | 0.0707 ± 0.0115(+) | 0.0350 ± 0.0032(+) |
| MMF2 | 0.0241 ± 0.0134 | 0.0293 ± 0.0023(+) | **0.0166 ± 0.0022**(≈) | 0.0245 ± 0.0043 (+) | 0.0511 ± 0.0167 (+) | 0.0705 ± 0.0518(+) | 0.0550 ± 0.0309(+) | 0.0505 ± 0.0248(+) |
| MMF4 | **0.0135 ± 0.0008** | 0.0155 ± 0.0009(+) | **0.0135 ± 0.0009**(≈) | 0.0158 ± 0.0005(+) | 0.0424 ± 0.0072(+) | 0.0486 ± 0.0123(+) | 0.0853 ± 0.0258(+) | 0.0297 ± 0.0113(+) |
| MMF5 | **0.0445 ± 0.0013** | 0.0512 ± 0.0032(+) | 0.0456 ± 0.0018(≈) | 0.0540 ± 0.0031(+) | 0.1183 ± 0.0134(+) | 0.1141 ± 0.0104(+) | 0.1414 ± 0.0176(+) | 0.0673 ± 0.0090(+) |
| MMF7 | 0.0154 ± 0.0025 | 0.0155 ± 0.0005(≈) | **0.0124 ± 0.0004**(−) | 0.0158 ± 0.0006(≈) | 0.0266 ± 0.0060(+) | 0.0242 ± 0.0043(+) | 0.0469 ± 0.0078(+) | 0.0214 ± 0.0060(+) |
| MMF8 | 0.0345 ± 0.0019 | 0.0491 ± 0.0175(+) | **0.0330 ± 0.0017**(≈) | 0.0410 ± 0.0024(+) | 0.1184 ± 0.0380(+) | 0.1269 ± 0.0388(+) | 0.9033 ± 0.6981(+) | 0.7683 ± 0.5483(+) |
| MMF10 | 0.0931 ± 0.1445 | **0.0025 ± 0.0002**(−) | 0.0045 ± 0.0023(≈) | 0.0235 ± 0.0050(≈) | 0.1936 ± 0.1208(+) | 0.1145 ± 0.1244(≈) | 0.1550 ± 0.1401(+) | 0.0766 ± 0.1245(+) |
| MMF11 | 0.0042 ± 0.0003 | 0.0042 ± 0.0009(+) | 0.0038 ± 0.0002(−) | 0.0054 ± 0.0003(+) | 0.0045 ± 0.0003(≈) | 0.0043 ± 0.0002(≈) | 0.0034 ± 0.0004(−) | **0.0033 ± 0.0006**(−) |
| MMF12 | 0.0018 ± 0.0001 | 0.0039 ± 0.0005(+) | 0.0023 ± 0.0004(+) | 0.0041 ± 0.0006(+) | 0.0022 ± 0.0002(+) | 0.0075 ± 0.0124(+) | 0.0018 ± 0.0004(≈) | **0.0017 ± 0.0004**(≈) |
| MMF13 | **0.0270 ± 0.0010** | 0.0320 ± 0.0027(+) | 0.0294 ± 0.0014(+) | 0.0353 ± 0.0011(+) | 0.0694 ± 0.0079(+) | 0.0685 ± 0.0099(+) | 0.0939 ± 0.0307(+) | 0.0976 ± 0.0360(+) |
| MMF14 | 0.0465 ± 0.0022 | **0.0455 ± 0.0063**(+) | 0.0473 ± 0.0017(≈) | 0.0463 ± 0.0015(≈) | 0.0882 ± 0.0097(+) | 0.0798 ± 0.0039(+) | 0.0949 ± 0.0081(+) | 0.2313 ± 0.0503(+) |
| MMF15 | 0.0508 ± 0.0014 | **0.0482 ± 0.0017**(≈) | 0.0497 ± 0.0009(≈) | 0.0519 ± 0.0025(≈) | 0.0797 ± 0.0055(+) | 0.0674 ± 0.0043(+) | 0.0794 ± 0.0134(+) | 0.1030 ± 0.0180(+) |
| MMF1_e | 0.7103 ± 0.4414 | 0.4850 ± 0.0426(−) | **0.2863 ± 0.0485**(−) | 0.3527 ± 0.1149(≈) | 0.7098 ± 0.1579(≈) | 1.4092 ± 0.7136(+) | 1.1505 ± 0.4701(+) | 1.9979 ± 0.5509(+) |
| MMF14_a | 0.0531 ± 0.0019 | 0.0546 ± 0.0018(+) | **0.0517 ± 0.0016**(≈) | 0.0526 ± 0.0014(≈) | 0.1057 ± 0.0051(+) | 0.0977 ± 0.0045(+) | 0.1188 ± 0.0133(+) | 0.2744 ± 0.0857(+) |
| MMF15_a | 0.0596 ± 0.0034 | 0.0573 ± 0.0032(≈) | **0.0547 ± 0.0020**(−) | 0.0558 ± 0.0014(−) | 0.1073 ± 0.0092(+) | 0.0910 ± 0.0081(+) | 0.1053 ± 0.0121(+) | 0.1675 ± 0.0285(+) |
| MMF10_*l* | **0.0629 ± 0.0377** | 0.1820 ± 0.0168(+) | 0.1700 ± 0.0162(+) | 0.1663 ± 0.0023(+) | 0.1767 ± 0.0335(+) | 0.1622 ± 0.0386(+) | 0.1931 ± 0.0370(+) | 0.2046 ± 0.0058(+) |
| MMF11_*l* | **0.0690 ± 0.0452** | 0.2207 ± 0.0270(+) | 0.2407 ± 0.0178(+) | 0.2058 ± 0.0253(+) | 0.2501 ± 0.0002(+) | 0.2502 ± 0.0002(+) | 0.2507 ± 0.0005(+) | 0.2508 ± 0.0005(+) |
| MMF12_*l* | **0.0203 ± 0.0144** | 0.2602 ± 0.0354(+) | 0.2282 ± 0.0334(+) | 0.2122 ± 0.0397(+) | 0.2464 ± 0.0004(+) | 0.2462 ± 0.0002(+) | 0.2470 ± 0.0004(+) | 0.2453 ± 0.0028(+) |
| MMF13_*l* | **0.0817 ± 0.0119** | 0.2270 ± 0.0144(+) | 0.2485 ± 0.0112(+) | 0.2330 ± 0.0157(+) | 0.2798 ± 0.0093(+) | 0.2765 ± 0.0058(+) | 0.2971 ± 0.0144(+) | 0.3035 ± 0.0195(+) |
| MMF15_*l* | **0.0822 ± 0.0075** | 0.1506 ± 0.0156(+) | 0.1534 ± 0.0211(+) | 0.1468 ± 0.0080(+) | 0.2341 ± 0.0284(+) | 0.2451 ± 0.0197(+) | 0.2594 ± 0.0168(+) | 0.2712 ± 0.0196(+) |
| MMF15_a_*l* | **0.1021 ± 0.0120** | 0.1628 ± 0.0106(+) | 0.1605 ± 0.0129(+) | 0.1592 ± 0.0080(+) | 0.2037 ± 0.0187(+) | 0.2138 ± 0.0133(+) | 0.2288 ± 0.0104(+) | 0.2377 ± 0.0169(+) |
| MMF16_*l* 1 | **0.0755 ± 0.0060** | 0.1156 ± 0.0078(+) | 0.1101 ± 0.0083(+) | 0.1034 ± 0.0063(+) | 0.1725 ± 0.0064(+) | 0.1696 ± 0.0076(+) | 0.1767 ± 0.0074(+) | 0.2030 ± 0.0130(+) |
| MMF16_*l* 2 | **0.1018 ± 0.0123** | 0.1903 ± 0.0266(+) | 0.2035 ± 0.0307(+) | 0.1852 ± 0.0131(+) | 0.3129 ± 0.0183(+) | 0.3108 ± 0.0170(+) | 0.3273 ± 0.0235(+) | 0.3171 ± 0.0318(+) |
| MMF16_*l* 3 | **0.0885 ± 0.0073** | 0.1408 ± 0.0073(+) | 0.1462 ± 0.0128(+) | 0.1390 ± 0.0130(+) | 0.2143 ± 0.0113(+) | 0.2123 ± 0.0060(+) | 0.2252 ± 0.0104(+) | 0.2322 ± 0.0227(+) |
| + | | 19/24 | 11/24 | 17/24 | 22/24 | 22/24 | 22/24 | 22/24 |
| ≈ | | 3/24 | 8/24 | 6/24 | 2/24 | 2/24 | 1/24 | 1/24 |
| − | | 2/24 | 5/24 | 1/24 | 0/24 | 0/24 | 1/24 | 1/24 |

**Notes.**

Bold numbers indicate the best result in the row of data.

problems. In particular, MMODE_AP wins 12 instances over 24 test suites. From the tables, MMODE_AP and MO_PSO_MM algorithms have demonstrated significant superiority over other existing algorithms, whose highlighted mean value is the most. MMODE_AP obtained one order of magnitude better than the traditional migration algorithm in terms of IGDF in solving most MMOPLs (MMOP with a local Pareto front). DN-NSGAII and Omni-optimizer performed poorly on any of the 24 benchmark problems, although they are specially designed for MMOPs. Table 5 lists the mean and standard deviation IGDF results of the competitor algorithms. Intuitively, MMODE_AP wins 18 instances over 24 test problems. MMODE_AP has a great superior performance over competitors on the benchmarks. Note that for some instances, such as MMF5, MMF8, MMF10, MMF15 and MMF15_a, there is no substantial variation between the MMODE_AP and winner algorithms. Similar to that of IGDX, MMODE_AP obtained one order of magnitude better than the traditional migration algorithm in terms of IGDF in solving most MMOPLs. Tables 3–5 shows that MMODE_AP performs similarly to competitor algorithms for

**Table 5  Comparison of IGDF mean and standard deviation obtained by all algorithms for CEC 2020 benchmark problems.**

| IGDF | MMODE_AP | DSC-MOAGDE | MO_PSO_MM | MO_Ring_PSO_SCD | DN-NSGAII | Omni_Opt | NSGAII | SPEA2 |
|---|---|---|---|---|---|---|---|---|
| MMF1 | **0.0014 ± 0.0001** | 0.0019 ± 0.0001(+) | 0.0015 ± 0.0000(+) | 0.0021 ± 0.0001(+) | 0.0020 ± 0.0002(+) | 0.0017 ± 0.0002(+) | 0.0015 ± 0.0002(≈) | 0.0017 ± 0.0001(+) |
| MMF2 | **0.0039 ± 0.0004** | 0.0136 ± 0.0018(+) | 0.0092 ± 0.0009(+) | 0.0137 ± 0.0017(+) | 0.0124 ± 0.0106(+) | 0.0056 ± 0.0034(≈) | 0.0063 ± 0.0045(≈) | 0.0172 ± 0.0060(+) |
| MMF4 | **0.0012 ± 0.0001** | 0.0018 ± 0.0001(+) | 0.0014 ± 0.0001(+) | 0.0018 ± 0.0001(+) | 0.0016 ± 0.0001(+) | 0.0014 ± 0.0001(+) | **0.0012 ± 0.0001**(≈) | 0.0017 ± 0.0001(+) |
| MMF5 | 0.0014 ± 0.0001 | 0.0018 ± 0.0001(+) | 0.0015 ± 0.0000(≈) | 0.0020 ± 0.0001(+) | 0.0018 ± 0.0001(+) | 0.0015 ± 0.0002(≈) | **0.0013 ± 0.0001**(−) | 0.0017 ± 0.0001(+) |
| MMF7 | **0.0012 ± 0.0000** | 0.0018 ± 0.0009(+) | 0.0013 ± 0.0000(+) | 0.0018 ± 0.0001(+) | 0.0019 ± 0.0001(+) | 0.0015 ± 0.0000(+) | 0.0013 ± 0.0001(+) | 0.0017 ± 0.0001(+) |
| MMF8 | 0.0025 ± 0.0003 | 0.0028 ± 0.0003(+) | 0.0022 ± 0.0001(−) | 0.0029 ± 0.0002(+) | 0.0019 ± 0.0002(−) | 0.0016 ± 0.0001(−) | **0.0013 ± 0.0000**(−) | 0.0017 ± 0.0001(−) |
| MMF10 | 0.0917 ± 0.1209 | 0.0973 ± 0.0152(≈) | **0.0232 ± 0.0073**(≈) | 0.0938 ± 0.0109(≈) | 0.2178 ± 0.0714(≈) | 0.1134 ± 0.1079(≈) | 0.1530 ± 0.1189(≈) | 0.0911 ± 0.1035(≈) |
| MMF11 | **0.0111 ± 0.0009** | 0.0162 ± 0.0010(+) | 0.0114 ± 0.0012(≈) | 0.0174 ± 0.0017(+) | 0.0138 ± 0.0018(+) | 0.0120 ± 0.0014(≈) | **0.0111 ± 0.0008**(≈) | 0.0136 ± 0.0011(+) |
| MMF12 | **0.0021 ± 0.0001** | 0.0039 ± 0.0007(+) | 0.0040 ± 0.0003(+) | 0.0071 ± 0.0007(+) | 0.0031 ± 0.0005(+) | 0.0078 ± 0.0127(+) | 0.0022 ± 0.0002(+) | 0.0027 ± 0.0002(+) |
| MMF13 | 0.0134 ± 0.0005 | 0.0275 ± 0.0032(+) | 0.0177 ± 0.0017(+) | 0.0312 ± 0.0032(+) | 0.0218 ± 0.0034(+) | 0.0158 ± 0.0015(+) | 0.0137 ± 0.0008(≈) | 0.0173 ± 0.0011(+) |
| MMF14 | **0.0634 ± 0.0013** | 0.0665 ± 0.0017(+) | 0.0667 ± 0.0011(+) | 0.0694 ± 0.0021(+) | 0.0958 ± 0.0040(+) | 0.0847 ± 0.0033(+) | 0.0976 ± 0.0032(+) | 0.1819 ± 0.0322(+) |
| MMF15 | 0.1019 ± 0.0032 | **0.0971 ± 0.0022**(−) | 0.1022 ± 0.0030(≈) | 0.1074 ± 0.0035(+) | 0.1606 ± 0.0083(+) | 0.1354 ± 0.0086(+) | 0.1445 ± 0.0162(+) | 0.2343 ± 0.0578(+) |
| MMF1_e | **0.0044 ± 0.0017** | 0.0088 ± 0.0009(+) | 0.0056 ± 0.0006(+) | 0.0074 ± 0.0008(+) | 0.0096 ± 0.0076(+) | 0.0087 ± 0.0073(≈) | 0.0071 ± 0.0039(≈) | 0.0121 ± 0.0097(+) |
| MMF14_a | **0.0627 ± 0.0017** | 0.0671 ± 0.0019(+) | 0.0649 ± 0.0016(+) | 0.0682 ± 0.0023(+) | 0.1056 ± 0.0069(+) | 0.0886 ± 0.0032(+) | 0.1040 ± 0.0096(+) | 0.2093 ± 0.0214(+) |
| MMF15_a | 0.1051 ± 0.0050 | 0.1006 ± 0.0053(≈) | **0.0996 ± 0.0038**(−) | 0.1058 ± 0.0037(≈) | 0.1766 ± 0.0102(+) | 0.1492 ± 0.0118(+) | 0.1566 ± 0.0106(+) | 0.2747 ± 0.0347(+) |
| MMF10_l | **0.0752 ± 0.0342** | 0.1902 ± 0.0091(+) | 0.1596 ± 0.0150(+) | 0.1802 ± 0.0121(+) | 0.1938 ± 0.0206(+) | 0.1702 ± 0.0328(+) | 0.2002 ± 0.0401(+) | 0.2115 ± 0.0300(+) |
| MMF11_l | **0.0503 ± 0.0067** | 0.0906 ± 0.0082(+) | 0.0903 ± 0.0015(+) | 0.0803 ± 0.0054(+) | 0.0923 ± 0.0005(+) | 0.0919 ± 0.0005(+) | 0.0915 ± 0.0002(+) | 0.0932 ± 0.0006(+) |
| MMF12_l | **0.0166 ± 0.0029** | 0.0729 ± 0.0096(+) | 0.0794 ± 0.0089(+) | 0.0629 ± 0.0117(+) | 0.0824 ± 0.0001(+) | 0.0822 ± 0.0001(+) | 0.0822 ± 0.0001(+) | 0.0772 ± 0.0128(+) |
| MMF13_l | **0.0354 ± 0.0085** | 0.1104 ± 0.0233(+) | 0.1138 ± 0.0275(+) | 0.0882 ± 0.0143(+) | 0.1482 ± 0.0032(+) | 0.1453 ± 0.0007(+) | 0.1443 ± 0.0006(+) | 0.1475 ± 0.0015(+) |
| MMF15_l | **0.1431 ± 0.0042** | 0.1721 ± 0.0037(+) | 0.1634 ± 0.0035(+) | 0.1668 ± 0.0042(+) | 0.2034 ± 0.0074(+) | 0.1918 ± 0.0072(+) | 0.2044 ± 0.0087(+) | 0.2691 ± 0.0160(+) |
| MMF15_a_l | **0.1475 ± 0.0048** | 0.1632 ± 0.0037(+) | 0.1631 ± 0.0026(+) | 0.1657 ± 0.0019(+) | 0.2162 ± 0.0067(+) | 0.1962 ± 0.0064(+) | 0.2060 ± 0.0089(+) | 0.2819 ± 0.0234(+) |
| MMF16_l 1 | **0.1122 ± 0.0030** | 0.1363 ± 0.0048(+) | 0.1269 ± 0.0019(+) | 0.1289 ± 0.0027(+) | 0.1581 ± 0.0044(+) | 0.1529 ± 0.0034(+) | 0.1620 ± 0.0077(+) | 0.2411 ± 0.0183(+) |
| MMF16_l 2 | **0.1562 ± 0.0078** | 0.1996 ± 0.0055(+) | 0.1992 ± 0.0040(+) | 0.1992 ± 0.0033(+) | 0.2359 ± 0.0045(+) | 0.2293 ± 0.0073(+) | 0.2465 ± 0.0080(+) | 0.2933 ± 0.0230(+) |
| MMF16_l 3 | **0.1296 ± 0.0028** | 0.1597 ± 0.0032(+) | 0.1616 ± 0.0025(+) | 0.1590 ± 0.0033(+) | 0.1892 ± 0.0045(+) | 0.1850 ± 0.0058(+) | 0.1981 ± 0.0076(+) | 0.2549 ± 0.0168(+) |
| + | | 21/24 | 18/24 | 22/24 | 22/24 | 18/24 | 15/24 | 22/24 |
| ≈ | | 2/24 | 4/24 | 2/24 | 1/24 | 5/24 | 7/24 | 1/24 |
| − | | 1/24 | 2/24 | 0/24 | 1/24 | 1/24 | 2/24 | 1/24 |

**Notes.**

Bold numbers indicate the best result in the row of data.

normal MMOPs but significantly better for MMOPLs. In summary, MMODE_AP is competitive for MMOPLs and competent for normal MMOPs. Moreover, the Friedman test results are shown in Table 6, which shows that MMODE_AP obtains the best result on rpsp, IGDX and IGDF indicators. In terms of 1/HV, NSGAII achieved a better ranking than MO_Ring_PSO_SCD and SPEA2. This finding makes the 1/HV metric controversial as studies have reported MO_Ring_PSO_SCD and SPEA2 to be more comparative than NSGAII (*Shang et al., 2020*; *While et al., 2006*). Moreover, the 1/PSP, IGDX, and IGDF metrics give more consistent results than the 1/HV metric. In summary, comprehensive results reveal that the best comprehensive result is achieved by MO_PSO_MM, followed by MMODE_AP and DSC-MOAGDE. This indicated that the top three algorithms performed competitively.

Now, we discuss the performance of MMODE_AP. The most outstanding difference occurs with the results for MMF15_l on three objectives. Figs. S2 and S3 shows the

**Table 6   The Friedman test results including the score and ranking of each algorithm on four performance metrics.**

| Algorithm | 1/HV Score (ranking) | rpsp Score (ranking) | IGDX Score (ranking) | IGDF Score (ranking) | Comprehensive result Score (ranking) |
|---|---|---|---|---|---|
| MMODE_AP | 4.7083(6) | 2.0833(1) | 2.0417(1) | 1.4583(1) | 2.2500(2) |
| DSC-MOAGDE | 3.4167(1) | 3.3333(4) | 3.4583(4) | 4.5417(3) | 3.0000(3) |
| MO_PSO_MM | 3.6250(2) | 2.4583(2) | 2.5833(2) | 2.9583(2) | 2.0000(1) |
| MO_Ring_PSO_SCD | 5.9167(7) | 3.1250(3) | 3.1667(3) | 4.9167(5) | 4.7500(5) |
| DN-NSGAII | 4.0833(5) | 5.8750(5) | 5.9583(6) | 6.4167(7) | 6.0000(6) |
| Omni_Opt | 3.7917(3) | 5.9167(6) | 5.6667(5) | 4.6667(4) | 4.5000(4) |
| NSGAII | 3.9583(4) | 6.8333(8) | 6.7500(8) | 4.6667(4) | 6.2500(7) |
| SPEA2 | 6.5000(8) | 6.3750(7) | 6.3750(7) | 6.3750(6) | 7.2500(8) |

**Table 7   Comparison results of four indicators' mean value of MMF15_l benchmark.**

| MMF15_*l* | MMODE_AP | DSC-MOAGD | MO_PSO_MM | MO_Ring_PSO_SCD | DN_NSGAII | Omni_Opt | NSGAII | SPEA2 |
|---|---|---|---|---|---|---|---|---|
| rHV | 0.2666 | 0.2457 | 0.242 | 0.2389 | **0.2293** | 0.2305 | 0.2333 | 0.2368 |
| rPSP | **0.0892** | 0.1537 | 0.1441 | 0.1377 | 0.3698 | 0.3664 | 0.4705 | 0.4411 |
| IGDX | **0.0892** | 0.1537 | 0.1441 | 0.1377 | 0.2429 | 0.2474 | 0.2507 | 0.2808 |
| IGDF | **0.1478** | 0.1641 | 0.163 | 0.1662 | 0.2085 | 0.1959 | 0.2057 | 0.2689 |

**Notes.**
Bold numbers indicate the best result in the row of data.

distribution of the obtained solutions in the OS and DS on the MMF15_*l* benchmark, which has a global PF corresponding to one PS and a local PF corresponding to one PS. Notably, DSC-MOAGDE is considered the latest effective algorithm that can solve MMOP. As the figures illustrate, MMODE_AP shows significantly better performance compared with DSC-MOAGDE. Meanwhile, SPEA2 loses one PS and can find only a single non-uniform distribution PS. Table 7 shows the four indicators obtained by MMODE_AP on MMF15_*l* instances and the best values are highlighted in bold. From this table, we can observe that MMODE_AP performs best regarding rPSP, IGDX and IGDF indicators. However, other algorithms have worse rHV results than DN_NSGAII. It is reasonable to conclude that MMODE_AP locates all PSs with good convergence and overlap ratio, unlike most competitor algorithms that only detect one PS.

Due to article length limitations, the final approximated set for two (three) objectives of the test suite of CEC'2020 are not presented. For more information, please refer to Fig. S1. The visual comparisons show that the optimal solutions yielded by MMODE_AP successfully approximate the whole PFs. Especially for the top ten benchmarks, we can observe that the objective points generated by MMODE_AP distribute alone the PFs well on two- or three-dimensional space from Fig. S1. To summarize, MMODE_AP shows the superiority of balancing the convergence and diversity of solutions.

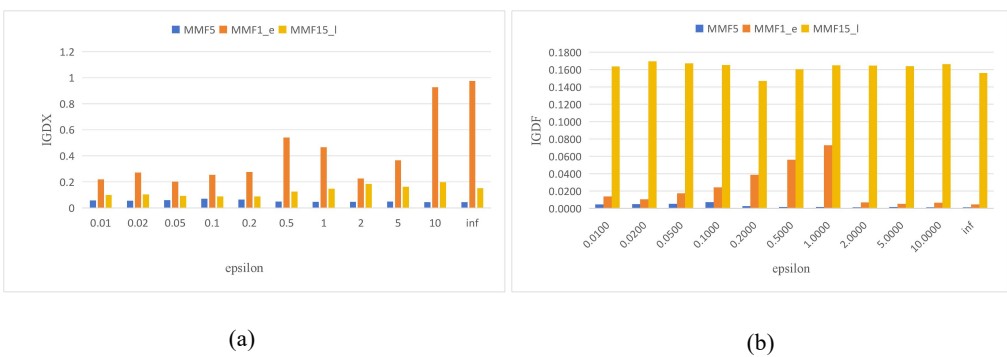

(a)                                                                                     (b)

**Figure 5  The IGDX and IGDF of MMODE_AP with different epsilon.** (A) IGDX. (B) IGDF.

## Parameter analysis

In the ***Archive update*** strategy, $\varepsilon$ is an important parameter that may affect the results of MMODE_AP. Improper parameter settings may lead to significant performance deterioration. In this section, the effect of the parameter on the performance of MMODE_AP is analyzed by performing relevant experiments on three representative functions (MMF5, MMF1_e, and MMF15_*l*). First, $\varepsilon$ is empirically set as $\{0.01, 0.02, 0.05, 0.1, 0.2, 0.5, 1, 2, 5, 10, inf\}$. Figure 5 presents the average values calculated over the 3 test problems for each indicator (*i.e.,* IGDX and IGDF). From Fig. 5A, we can observe that a good average IGDX is obtained when the parameter value is in $[0.01, 0.2]$. For the MMF1_e test problem, the results are clearly bad when $\varepsilon$ is too large. In Fig. 5B, the IGDF value remains constant for the MMF5 and MMF15_*l* test problems as $\varepsilon$ increases. For the MMF1_e test problem, the IGDF values are fluctuating. Based on the above discussion, the performance of MMODE_AP is favorable when $\varepsilon \in [0.01, 0.1]$. However, limitations also exist in this experiment. For example, the parameter range for obtaining satisfactory results is wide. Therefore, it increases the cost of time for decision-makers.

## Analysis of the effectiveness of the proposed solution generation

To verify the effectiveness of the proposed selection mechanism, MMODE_AP with different $p_1$ are compared. The results on MMF3 are taken as an example and they are similar to the other test functions. The obtained PSs with median 1/PSP in the 21 times of these parameters are shown in Fig. 6. The Fig. 6A denotes the parameter $p_1$ selected a linear decreasing function. Figure 6B denotes the parameter $p_1$ selected a constant 1, which presents the DE/rand/2 strategy is adopted only. The Fig. 6C denotes the parameter $p_1$ selected a constant 0.5. Figure 6D denotes the parameter $p_1$ is a constant 0, which presents the DE/current-to-exemplar/1 strategy is adopted only. As shown in Figs. 6B–6D, the population doesn't converge to the true PSs well. The obtained PSs of MMF3 are complete as shown in Fig. 6A. Overall, the the proposed selection mechanism is effective.

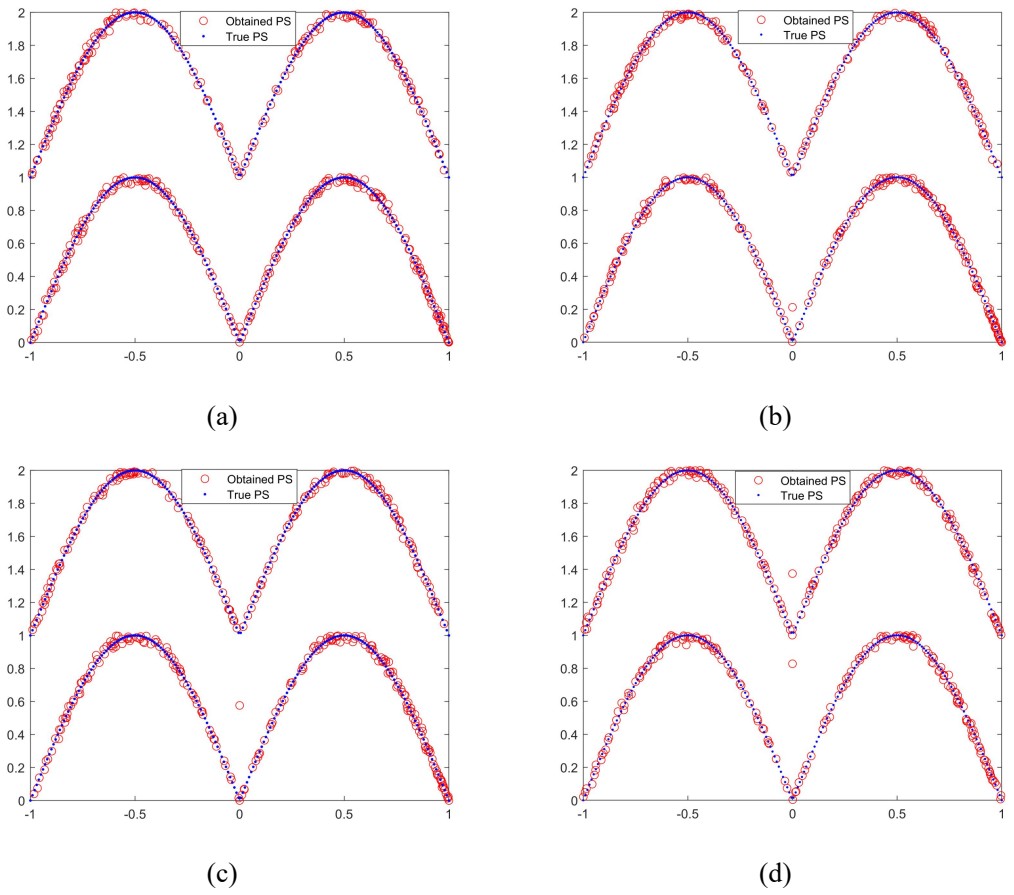

**Figure 6 The obtained PSs of MMODE_AP with different p1.** (A) p_1: a linear decreasing function. (B) $p_1 = 1$. (C) $p_1 = 0.5$. (D) $p_1 = 0$.

## CONCLUSION

In order to enhance the exploration and exploitation of MMODE, this article proposes an improved MMODE named MMODE_AP to solve MMOPs. In MMODE_AP, the affinity propagation clustering method groups individuals in non-dominated layers into different classes, allowing individuals and their neighbors to be grouped in the same Pareto set. Based on this method, the convergence of this algorithm can be accelerated in the process of optimization evolution. Then, two mutation strategies are adopted in the solution generation process. Based on the strategy, not only the exploration and exploitation performance are balanced but also the diversities in DS and OS are improved. Moreover, the CSCD is employed to create a promising generation in environmental selection. The comprehensive CD can reflect the actual crowding degree accurately. Additionally, an archive updating approach is designed to balance the convergence and diversity quality of solutions, in which a predefined parameter is designed to control the completeness of the local and local PFs. To demonstrate the potential benefit of the novel algorithm, a series

of comprehensive experiments on CEC'2020 multi-modal multi-objective benchmark instances are conducted. Here is a summary of the pros and cons of the suggested approach:

The proposed method has several advantages. Thanks to the implementation of the APC technique, the local and global PS (PF) can be well maintained. However, some limitations are also found in the experiment. For example, the performance of MMODE_AP in OS needs further improvements as it loses to competitors in terms of HV metric. Except for benchmark suites, developing potential multi-objective real-world test problems that contain multi-modal challenges for researchers is also essential. In future work, the implementation of the algorithm is to be discussed, including the potential for parallelization and optimization. Furthermore, more efficient maintenance mechanisms of local PSs need to be studied.

### Funding
This work was supported by the Opening Project of Sichuan Province University Key Laboratory of Bridge Non-destruction Detecting and Engineering Computing (2021QYY03, 2021QYY02, 2020QYY02). The funders had no role in study design, data collection and analysis, decision to publish, or preparation of the manuscript.

### Grant Disclosures
The following grant information was disclosed by the authors:
Opening Project of Sichuan Province University Key Laboratory of Bridge Non-destruction Detecting and Engineering Computing: 2021QYY03, 2021QYY02, 2020QYY02.

### Competing Interests
The authors declare there are no competing interests.

### Author Contributions
- Dan Qu conceived and designed the experiments, authored or reviewed drafts of the article, and approved the final draft.
- Hualin Xiao performed the experiments, prepared figures and/or tables, and approved the final draft.
- Huafei Chen performed the computation work, authored or reviewed drafts of the article, and approved the final draft.
- Hongyi Li analyzed the data, prepared figures and/or tables, and approved the final draft.

### Data Availability
The raw measurements are available in the Supplementary Files.
The CEC2020 benchmark functions are available at GitHub: https://github.com/P-N-Suganthan/2020-Multimodal-Multi-Objective-Benchmark.

## Supplemental Information

Supplemental information for this article can be found online at http://dx.doi.org/10.7717/ peerj-cs.1839#supplemental-information.

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
