# Peer review of "An improved differential evolution algorithm for multi-modal multi-objective optimization"

_PeerJ Computer Science, doi:10.7717/peerj-cs.1839_

## Round 0.1 · original submission · Major Revisions

The two reviewers agree that the idea is interesting, but they both highlight a number of shortcomings with the presentation (of the experimental setup in particular) and with the figures. The second reviewer offers a number of great suggestions in this respect.

Also, could you please address the main concern of Reviewer 1 by providing all implementations used for the experiments and referring to them appropriately in the manuscript? I note that the CEC' 2020 dataset has been submitted, but having more details will help readers who want to reproduce the results in the paper.

Reviewer 1 ·

Basic reporting

The paper is interesting, however, the method and results can be improved, the method is unambiguous and the Figures are not clear. Also, the information presented in tables can be transform to figures. the method should include clear definitions, the algorithms are confused and the benchmark should be more described.

Experimental design

The experimental design is based on benchmark, it should be described better.

Validity of the findings

All underlying data and code have not provided

Reviewer 2 ·

Basic reporting

The manuscript presents a modification of the differential evolution for solution of the multiomodal multiobjective optimization problems. The method is able to handle box constraints on the parameters. Two mutation strategies are used with a probability that depends on the iteration to enhance the exploration and the exploitation phases. The affinity propagation clustering and crowding distance are adopted to find both local and global Pareto sets. The method was evaluated using the functions from CEC2020 benchmark set and showed promising results.

The approach proposed by authors is very interesting and the combination of the algorithms seems to be really powerful.

The overall English language quality is good and the paper is easy to follow. Some phrases might be enhanced though.

The quality and clarity of figures is to be enhanced.

Details are in 4. Additional comments.

Experimental design

The research is in the scope of the journal, the problem is well defined and meaningfull. The numerical experiments are performed with a well established procedure. The description of methods is to be enhanced. See details in 4. Additional comments.

Validity of the findings

The impact and the limitation of the approach are to be discussed in more details. the conclusions are to be strengthen. See details in 4. Additional comments.

Additional comments

I suggest to revise the text to address the following comments before publication.

Some numerical results is to be included in the abstract.

line 42 What is f without a subscript?

Please state more clearly why the solutions of exisiting methods are not enough for practical MMOPs and obtaining more points in DS is important.

Please decode all abbreviations on the first use including methods' names in Related studies.

lines 200-210 The DE was also successfully applied in the area of computational biology which is worth to be mentioned here as
The references from this paragraph 51-56 are rather old - 2003-2012. Please include more recent applications as they are numerous in different fields including computational biology.

Algorithm 3. Please give the details for p.2.1 - how matrices A and R are calculated. In step 2.2 the space is in the wrong place: "tot he" is to be "to the". How is the number of clusters determined?

line 244, Fig.3 The font in the figure is to be made bigger relative to the boxes, the notation is to be made less confusing: now the elements of matrix P are called x while the elements of matrix F are called p.

Algorithm 4 step 5 - please introduce P*, it was not mentioned before.

Algorithm 5. Please explain in more detail what is x_{exemplar} and how it is obtained or selected. Please clarify the crossover in step 4, the meaning of j and k. What distribution is used?

line 259. Please justify the expression for p1. One of the advantages of DE is that there is no "temperature" and "cooling schedule" so the inclusion of such step needs to be proven. Also the "cooling schedule" itself may be the topic of the research.

line 263. Please explain why this method is used to fullfill the constraints on the parameters. There are other approaches that may be more convinient for DE, see e.g.

Cantú, Victor H., Catherine Azzaro-Pantel, and Antonin Ponsich. 2021. “Constraint-Handling Techniques within Differential Evolution for Solving Process Engineering Problems.” Applied Soft Computing 108 (September): 107442. https://doi.org/10.1016/j.asoc.2021.107442.

Kozlov, Konstantin, Alexander M. Samsonov, and Maria Samsonova. 2016. “A Software for Parameter Optimization with Differential Evolution Entirely Parallel Method.” PeerJ Computer Science 2 (August): e74. https://doi.org/10.7717/peerj-cs.74.

Lampinen, J. 2002. “A Constraint Handling Approach for the Differential Evolution Algorithm.” In Proceedings of the 2002 Congress on Evolutionary Computation. CEC’02 (Cat. No.02TH8600), 2:1468–73. Honolulu, HI, USA: IEEE. https://doi.org/10.1109/CEC.2002.1004459.

line 316 Why Adis in eq.2 is an average?

line 340 Please describe in more detail the main purpose of the Archive. In some modifications of DE archive is used to store the promising directions for moves in parameter states, see e.g.

Ghosh, Arka, Swagatam Das, Asit Kr. Das, and Liang Gao. 2020. “Reusing the Past Difference Vectors in Differential Evolution—A Simple But Significant Improvement.” IEEE Transactions on Cybernetics 50 (11): 4821–34. https://doi.org/10.1109/TCYB.2019.2921602.

Please, specify the axes in Fig.5. The individual panels are rather small and might be enlarged as there is empty space in the page.

line 441 "Good convergence and overlap ratio" as well as "approximation" of the whole PFs (line 448) are to be quantified and properly quantitatively compared.

The implementation of the algorithm is to be discussed including the potential for parallelization and optimization.

It would be useful if the authors provide the complete list of the hyperparameters of the developed algorithm together with the values used in the experiments and reasons of the selection.

The discussion of the results of the numerical experiments and overall results of the work is to be enlarged in the form of the separate section or combined with section 4. This should include the limitations that are now just mentioned in the Conclusions.

It would be great if the authors could draw stronger conclusions than just stating the advantages.

---

## Round 0.2 · Minor Revisions

You will see that the two reviewers recommend a number of minor changes to the manuscript.

Reviewer 1 ·

Basic reporting

The article's structure should adhere to an acceptable format. It is advisable to review the tables as they appear to be unreliable, causing the information to be unclear.


The figures ought to be pertinent to the article's content, possess adequate resolution, and be accurately described and labeled. Additionally, it's crucial to note that the figures are quite small, and some of the schematics may not be necessary.

Experimental design

ok

Validity of the findings

ok

Reviewer 2 ·

Basic reporting

The authors reworked many parts of the manuscript and substantially improved it. However some points still need attention in my opinion. See Additional comments section.

Experimental design

The authors reworked many parts of the manuscript and substantially improved it. However some points still need attention in my opinion. See Additional comments section.

Validity of the findings

The authors reworked many parts of the manuscript and substantially improved it. However some points still need attention in my opinion. See Additional comments section.

Additional comments

The authors responded that added numerical results to the abstract but I don't see those. For example, the sentence "Experimental results and statistical analysis reveal that MMODE_AP significantly outperforms the compared algorithms." can be modified to include the number of benchmarks or indicators in which new method outperforms competitors and the best result, or something like this.

The description of the algorithm was well clarified but I would like to comment the formula for p_1 that was mentioned in the previous report in comment 16. The terms temperature and cooling schedule came from the method of simulated annealing (SA) that uses the physical metaphor for optimization. The size of parameter changes during the optimization process is controlled by the temperature that is lowered from iteration to iteration according to cooling schedule. Numerous schedules were designed with different properties and the success of the methods greatly relays on them. The parameter p_1 serves the same goal as it controls the probability of far moves that is lowered with each generation. There are several DE variants with similar parameters as the idea is rather straightforward. Consequently my original comment from report 1 was based on that the original DE doesn't have such a parameter and in many cases it may be considered an advantage and thus introduction of such parameter is to be justified together with the formula for p_1 as there are other decreasing functions that can be used here similar to SA cooling schedules.

Some language issues remain, e.g.

line 49 The set of all non-dominated solutions consists of PS -> The set of all non-dominated solutions constitutes PS

line 295 the random value is less than p_1

---

## Round 0.3 · Minor Revisions

As you will see, one of the reviewer has identified one issue that needs addressing. Please address this well-defined query and update the manuscript accordingly.

Reviewer 2 ·

Basic reporting

The authors took into account almost all my suggestions. The details are in Additional comments.

Experimental design

The authors took into account almost all my suggestions. The details are in Additional comments.

Validity of the findings

The authors took into account almost all my suggestions. The details are in Additional comments.

Additional comments

Dear Authors,

May be my comment about p_1 in the previous report was not clear - my apologies.

The text in lines 262-266 in the current version of the manuscript is correct but it doesn't answer my question. By the words 'introduction of such a parameter is to be justified' I meant that you need to explain why you added parameter p_1 and why you selected a linear decreasing function. Did you try without such a parameter, did you try different constants in place of it, etc.

---

## Round 0.4 · accepted · Accept

The reviewer has confirmed that all comments have been addressed satisfactorily. I am therefore happy to accept this manuscript for publication.

Reviewer 2 ·

Basic reporting

The authors took all my suggestions into account so I don't have any further comments and suggest to accept the manuscript

Experimental design

The authors took all my suggestions into account so I don't have any further comments and suggest to accept the manuscript

Validity of the findings

The authors took all my suggestions into account so I don't have any further comments and suggest to accept the manuscript